# Localized inhibition in the *Drosophila* mushroom body

**Hoger Amin[1,2], Anthi A Apostolopoulou[1,2], Raquel Suárez-Grimalt[1†], Eleftheria Vrontou[3], Andrew C Lin[1,2]\***

[1]Department of Biomedical Science, University of Sheffield, Sheffield, United Kingdom; [2]Neuroscience Institute, University of Sheffield, Sheffield, United Kingdom; [3]Centre for Neural Circuits and Behaviour, University of Oxford, Oxford, United Kingdom

**Abstract** Many neurons show compartmentalized activity, in which activity does not spread readily across the cell, allowing input and output to occur locally. However, the functional implications of compartmentalized activity for the wider neural circuit are often unclear. We addressed this problem in the *Drosophila* mushroom body, whose principal neurons, Kenyon cells, receive feedback inhibition from a non-spiking interneuron called the anterior paired lateral (APL) neuron. We used local stimulation and volumetric calcium imaging to show that APL inhibits Kenyon cells' dendrites and axons, and that both activity in APL and APL's inhibitory effect on Kenyon cells are spatially localized (the latter somewhat less so), allowing APL to differentially inhibit different mushroom body compartments. Applying these results to the *Drosophila* hemibrain connectome predicts that individual Kenyon cells inhibit themselves via APL more strongly than they inhibit other individual Kenyon cells. These findings reveal how cellular physiology and detailed network anatomy can combine to influence circuit function.

**\*For correspondence:**
andrew.lin@sheffield.ac.uk

**Present address:** [†]Institute of Neurophysiology, Charité - Universitätsmedizin Berlin, Berlin, Germany

**Competing interests:** The authors declare that no competing interests exist.

## Introduction

A textbook neuron integrates input from dendrites to generate action potentials that spread throughout the neuron, making it a single functional unit. However, many neurons integrate inputs and generate outputs locally so that a single neuron effectively functions as multiple independent units (reviewed in *Branco and Häusser, 2010*). What functional consequences flow from such compartmentalized activity? In some cases, the circuit function of compartmentalized activity is clear (*Euler et al., 2002*; *Grimes et al., 2010*; *Meier and Borst, 2019*), but answering this question is often difficult due to the complexity of mammalian nervous systems or lack of detailed anatomical information.

These difficulties are eased in the *Drosophila* olfactory system, a numerically simple circuit that is now subject to intensive connectomic reconstruction (*Takemura et al., 2017*; *Tobin et al., 2017*; *Scheffer et al., 2020*; *Zhang et al., 2019*; *Zheng et al., 2018*), offering an excellent opportunity to study local computations. We focus on the mushroom body, the fly's olfactory learning center, which has a well-characterized compartmentalized architecture (reviewed in *Amin and Lin, 2019*). The mushroom body's principal neurons, the Kenyon cells (KCs), form parallel bundles of axons that are divided into compartments by the innervation patterns of dopaminergic neurons and mushroom body output neurons. During olfactory associative learning, dopaminergic neurons in one compartment modify the local connection strength between Kenyon cells and mushroom body output neurons in the same compartment, but not other compartments (*Cohn et al., 2015*; *Hige et al., 2015*).

Superimposed on this compartmentalized structure is an inhibitory interneuron that innervates all the mushroom body compartments: the anterior paired lateral (APL) neuron (*Aso et al., 2014*; *Tanaka et al., 2008*). The APL neuron ensures sparse odor coding by Kenyon cells through feedback

inhibition (*Lei et al., 2013*; *Lin et al., 2014*) and also plays a role in memory suppression and reversal learning (*Liu and Davis, 2009*; *Ren et al., 2012*; *Wu et al., 2012*; *Zhou et al., 2019*). APL's widespread innervation stands in contrast to the generally compartmentalized structure of the mushroom body, yet its diverse roles in memory hint that it may have a more compartmentalized function than its morphology suggests.

Indeed, there are hints that APL may have compartmentalized function. Like its locust homolog, the giant GABAergic neuron (GGN), APL is non-spiking (*Papadopoulou et al., 2011*). Unlike the larval APL (*Eichler et al., 2017*; *Masuda-Nakagawa et al., 2014*), the adult APL has pre- and post-synaptic specializations throughout all its processes (*Wu et al., 2013*), and connectomic reconstructions show that APL forms reciprocal synapses with Kenyon cells throughout the mushroom body (*Takemura et al., 2017*; *Scheffer et al., 2020*; *Zheng et al., 2018*). There have been reports of localized activity within APL (*Inada et al., 2017*; *Wang et al., 2019*), and compartmental modeling suggests that in the locust GGN, depolarization should diminish with distance from the site of current input, although not to zero (*Ray et al., 2020*). However, intracellular activity propagation is difficult to measure experimentally in locusts, and the lack of action potentials in APL at the soma (*Papadopoulou et al., 2011*) does not rule out active conductances elsewhere, given that mammalian dendrites often show dendritic spikes that do not necessarily propagate to the soma (*Branco and Häusser, 2010*). There have been no systematic studies of how widely activity propagates within APL, where it exerts its inhibitory effect on Kenyon cells, or how APL's physiology would affect the structure of feedback inhibition in the mushroom body.

We addressed these questions by volumetric calcium imaging in APL. By quantifying the spatial attenuation of the effects of locally stimulating APL, we found that both activity in APL and APL's inhibitory effect on Kenyon cells are spatially restricted (though the latter somewhat less so), allowing APL to differentially inhibit different compartments of the mushroom body. Combining these physiological findings with recent connectomic data predicts that individual Kenyon cells inhibit themselves via APL more strongly than they do other individual Kenyon cells. Our findings establish APL as a model system for studying local computations and their role in learning and memory.

## Results

### Spatially restricted responses to electric shock in APL

We first asked whether physiological responses to sensory stimuli are spatially localized within APL. At very low odor concentrations, odor responses in APL can be restricted to the β′ lobe of the horizontal lobe (*Inada et al., 2017*), but it remains unclear how localized sensory-evoked activity is across APL's whole innervation of the mushroom body lobes. We examined this question using electric shock, a typical 'punishment' used for olfactory aversive conditioning.

APL responds to electric shock (*Liu and Davis, 2009*; *Zhou et al., 2019*), but it is unknown if these responses are spatially restricted. To test this, we subjected flies to electric shock provided by copper plates touching the abdomen and legs, and recorded activity volumetrically throughout the APL lobes using GCaMP6f driven by 474-GAL4 (*Silies et al., 2013*). Although 474-GAL4 is not specific to APL, APL is most likely the only 474-GAL4+ neuron in the mushroom body, given that APL's typical morphology (neurites parallel to Kenyon cell axons) is visible throughout, without obvious innervation of other neurons (*Figure 1—figure supplement 1*). We divided the MB lobes into four regions: the tip of the vertical lobe (V), stalk of the vertical lobe (S), compartments γ1–3 (γ), and the rest of the horizontal lobe (H) (*Figure 1A*). We based these divisions on findings that (1) these regions are innervated by different dopaminergic neurons that respond differentially to 'punishment' (e.g. electric shock) vs. 'reward' (reviewed in *Amin and Lin, 2019*), (2) dopaminergic neurons release synaptic vesicles onto APL in response to electric shocks (*Zhou et al., 2019*), and (3) APL expresses the dopamine receptors DopEcR (*Aso et al., 2019*) and the inhibitory Dop2R (*Zhou et al., 2019*).

Electric shocks evoked a strong response in APL in the V region, with significantly smaller responses in the other regions (*Figure 1B,C*, left). This difference between regions did not arise from optical artifacts or differential expression of voltage-gated calcium channels. If it did, one would expect other stimuli to produce a similar pattern of $Ca^{2+}$ influx. However, the odor isoamyl acetate at $10^{-2}$ dilution evoked a different pattern of activity, with somewhat stronger responses in the S region and weaker in γ (*Figure 1B,C*, right). These different spatial profiles for electric shock

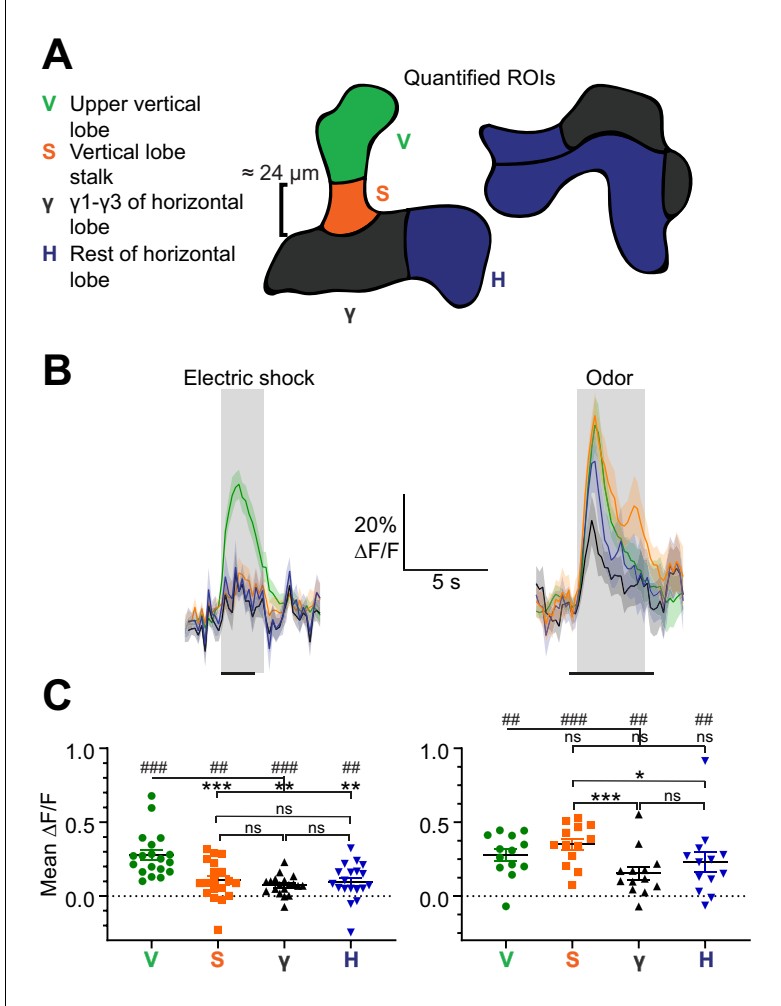

**Figure 1.** Spatially restricted responses to electric shock in APL. (A) Diagrams of regions used to quantify APL activity in (B–C). (B) APL responses to electric shock (left) or the odor isoamyl acetate (right) in the regions defined in (A), in 474-GAL4 > GCaMP6f flies. Traces show mean ± SEM shading. Electric shock trace is the average of 3 shocks with 12 s between onset of each shock. Horizontal black lines show timing of electric shock (1.2 s) or odor (5 s). (C) Mean ΔF/F during the intervals shaded in panel (B). ## p<0.01, ### p<0.001, one-sample Wilcoxon test vs. null hypothesis (0) with Holm-Bonferroni correction for multiple comparisons. *p<0.05, ***p<0.001, Friedman test with Dunn's multiple comparisons test for each group (V, H, S, and γ) against each other. n, given as # neurons (# flies): electric shock, 19 (15); odor, 13 (11).

The online version of this article includes the following source data and figure supplement(s) for figure 1:

**Source data 1.** Source data for *Figure 1C*.
**Figure supplement 1.** Expression pattern of 474-GAL4.

---

vs. odor suggest that spatially differential responses to electric shock might arise from different synaptic inputs. This could arise because different regions of APL (1) have dopamine receptors of different function (DopEcR vs. Dop2R), (2) receive dopaminergic inputs of different synaptic numbers, or (3) receive input from dopaminergic neurons with different responses to electric shock (*Cohn et al., 2015*; *Mao and Davis, 2009*). In support of (1), bath-applying dopamine inhibits APL in the vertical lobe, except in the tip where it has no direct effect (*Zhou et al., 2019*). In support of (2), APL receives more synapses from the dopaminergic neurons PPL1γ2α'1, PPL1γ1pedc and PPL1α'2α2 in the heel and stalk, which are presumably inhibitory (*Zhou et al., 2019*), compared to dopaminergic neurons PPL1α3 and PPL1α'3 in the tip (*Scheffer et al., 2020*). Regardless of the cause, the spatial heterogeneity of sensory-evoked activity within APL suggests that activity can remain spatially localized in APL.

# Low relative expression of voltage-gated Na⁺ and Ca²⁺ channel mRNAs in APL

In addition to reflecting differential synaptic inputs, this spatially localized activity in APL most likely also reflects intrinsic properties of APL's membrane excitability that prevent activity from spreading. To validate the hypothesis that intrinsic properties could contribute, we analyzed an RNA-seq data-set revealing mRNA levels in the cell bodies of Kenyon cells, dopaminergic neurons, mushroom body output neurons, the dorsal paired medial (DPM) neuron, and APL (*Aso et al., 2019*).

We looked for genes with higher or lower expression (transcripts per million) in APL than in all other 20 mushroom body neuron types. Gene Ontology analysis revealed that the set of genes with consistently higher expression in APL is strongly enriched for genes involved in protein synthesis and mitochondrial respiration, perhaps reflecting the metabolic demands of maintaining such a large neuron. APL also expresses higher levels of the biosynthetic enzyme and vesicular transporter of GABA than any other neuron in the dataset (*Figure 2*). In contrast, voltage-gated Na⁺ and Ca²⁺ channels are expressed at lower levels in APL than in all other types of mushroom body neurons (*Figure 2*). In contrast, the picture with K⁺ channels, ligand-gated Cl⁻ channels, and other miscellaneous

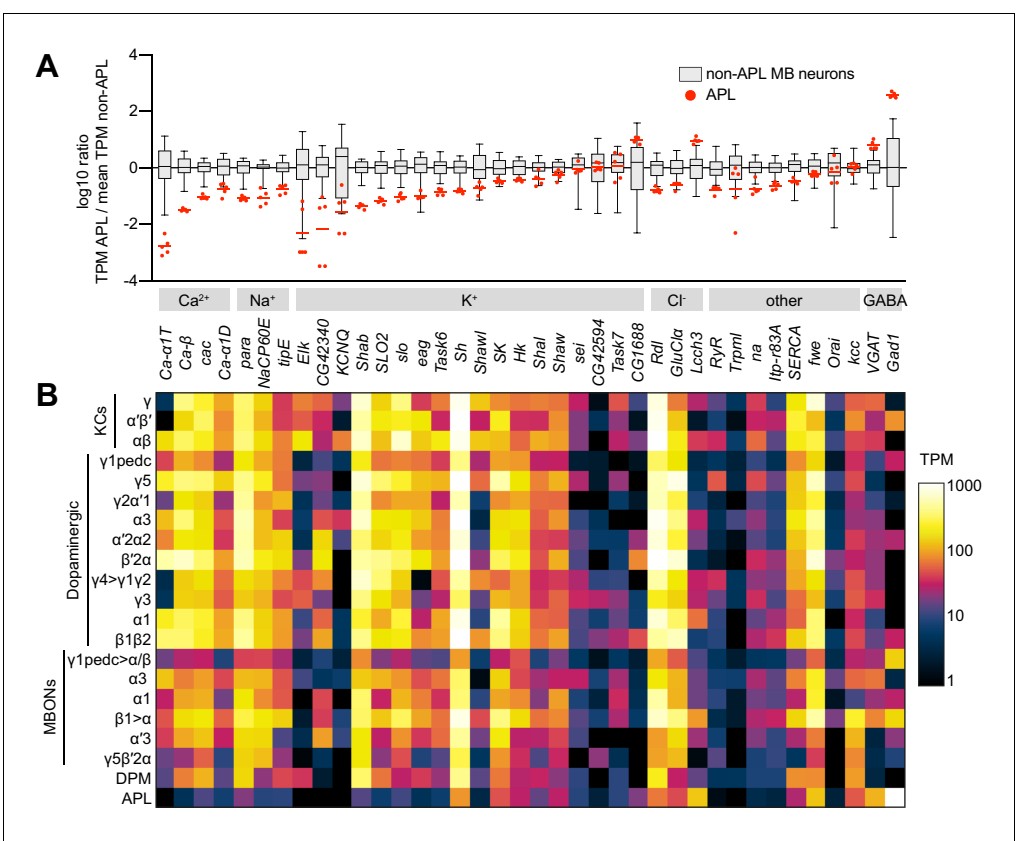

**Figure 2.** Low expression of voltage-gated Na⁺ and Ca²⁺ channels in APL. (**A**) Expression levels of GABAergic markers and various Na⁺, Ca²⁺, K⁺, Cl⁻ and other channels, in APL compared to non-APL mushroom body neurons, normalized to the mean of non-APL neurons. Box plots show the distribution of expression levels in non-APL MB neurons (whiskers show full range), using the average of log$_{10}$(TPM) across biological replicates for each type. Red dots show five biological replicates of APL; red line shows the mean APL expression level. Genes are grouped by type and sorted by fold difference in APL expression relative to non-APL MB neurons. Data from *Aso et al., 2019*. TPM, transcripts per million. (**B**) Heat map of TPM counts (log scale) for all genes shown in (**A**), for all 21 cell types.

The online version of this article includes the following source data and figure supplement(s) for figure 2:

**Source data 1.** Source data for *Figure 2*.

**Figure supplement 1.** When APL expresses Ort, histamine reduces odor-evoked Ca²⁺ influx in APL.

**Figure supplement 1—source data 1.** Source data for *Figure 2—figure supplement 1*.

channels is more mixed (*Figure 2*). These results are consistent with previous findings that APL does not fire action potentials (*Papadopoulou et al., 2011*), unlike all other published electrophysiological recordings from other mushroom body neurons (*Hige et al., 2015*; *Takemura et al., 2017*; *Turner et al., 2008*).

However, note that relative somatic mRNA expression (transcripts per million) may not necessarily translate into absolute protein abundance or ion conductance. Indeed, despite the relatively low expression of voltage-gated $Ca^{2+}$ channel mRNA, APL must have a significant voltage-gated $Ca^{2+}$ conductance. When the histamine-gated $Cl^-$ channel Ort (*Gengs et al., 2002*; *Liu and Wilson, 2013*) is ectopically expressed in APL, bath applying histamine strongly increases Kenyon cell odor responses (as strongly as blocking APL synaptic output with tetanus toxin); histamine has no effect when APL does not express Ort (*Apostolopoulou and Lin, 2020*). Such a result would be impossible if odor-evoked $Ca^{2+}$ influx (and hence GABA release) in APL were voltage-independent (i.e. if $Ca^{2+}$ entered entirely through nicotinic receptors or from internal stores), because Ort inhibits neurons by opening a $Cl^-$ conductance (*Liu and Wilson, 2013*; *Zheng et al., 2002*), not regulating $Ca^{2+}$, and could therefore only affect $Ca^{2+}$ influx via voltage. Indeed, when APL expresses Ort, histamine

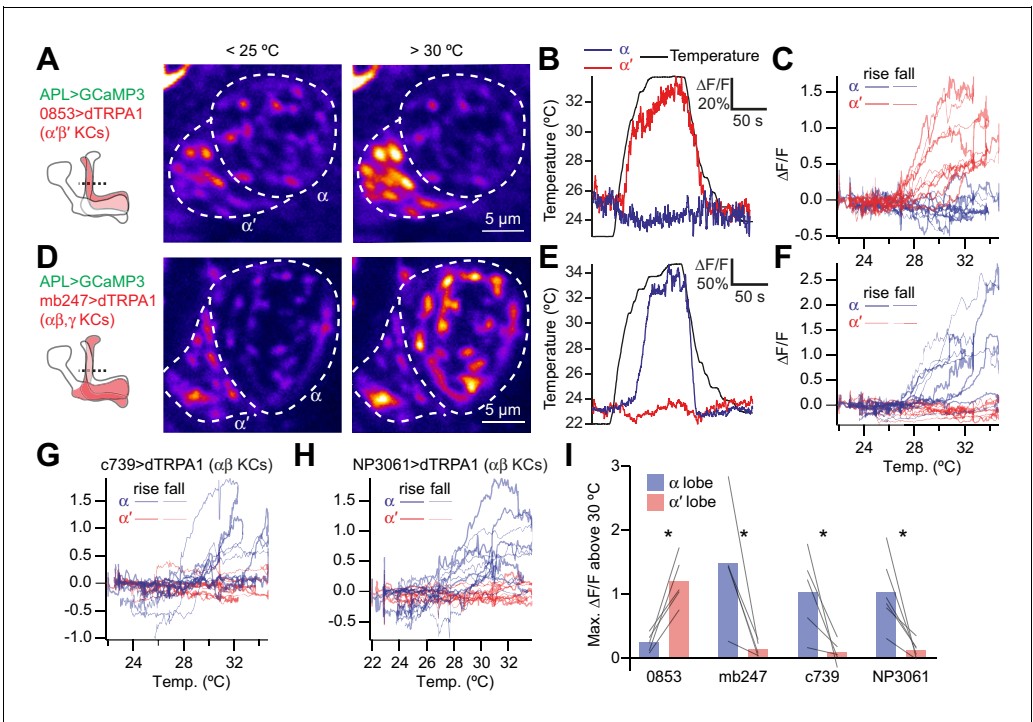

**Figure 3.** Activating subsets of Kenyon cells only activates the corresponding lobe of APL. (**A**) Time-averaged images of APL in a cross-section of the lower vertical lobe (dotted line in diagram) at room temperature (<25 ˚C) vs. heated (>30 ˚C). Activating α′β′ Kenyon cells expressing 0853-GAL4-driven dTRPA1 activates the α′ lobe, but not the α lobe, of APL. APL expresses GCaMP3 driven by GH146-QF. (**B**) Example responses to heating (black), by the α′ (red) and α (blue) lobes in the same APL recording. (**C**) ΔF/F for the α′ (red) and α (blue) lobes plotted against temperature. Each trace representing one recording forms a loop: it rises as the fly is heated (thick lines) and falls down a different path as the fly is cooled (thin lines). (**D–F**) As for (**A–C**) except activating αβ and γ Kenyon cells (mb247-GAL4 > dTRPA1) drives activity in the α, but not the α′ lobe of APL. (**G,H**) As for (**C**) and (**F**) except activating αβ Kenyon cells with c739-GAL4 (**G**) or NP3061-GAL4 (**H**). (**I**) Maximum ΔF/F between 30 ˚C and the peak temperature, summarizing data from (**C,F–H**). Lines indicate the same APL recording. *p<0.05, paired t-test with Holm-Bonferroni multiple comparisons correction. n (# flies), left to right: 5, 5, 5, 6. The expression patterns of mb247-GAL4, c739-GAL4, NP3061-GAL4 and GH146-QF were described previously (*Potter et al., 2010*; *Qin et al., 2012*; *Zhou et al., 2019*). The expression pattern of 853-GAL4 is available at http://www.columbia.edu/cu/insitedatabase/adultbrain/IT.0853.jpg.

The online version of this article includes the following source data for figure 3:

**Source data 1.** Source data for *Figure 3I*.

strongly reduces odor-evoked $Ca^{2+}$ influx in APL (*Figure 2—figure supplement 1*), indicating that $Ca^{2+}$ influx in APL is in large part voltage-dependent.

## Activity in APL is spatially restricted

Based on the spatially restricted sensory-evoked activity and the low (although non-zero) expression of voltage-gated ion channel mRNA, we hypothesized that intrinsic factors limit the spatial spread of activity within APL. We first tested this by imaging APL while activating Kenyon cells with the heat-activated cation channel dTRPA1. We previously showed that activating all Kenyon cells with dTRPA1 activates APL (*Lin et al., 2014*). We now extended this approach by activating only subsets of Kenyon cells. Kenyon cells are divided into three main classes, αβ, α′β′, and γ (see example morphologies in *Figure 8—figure supplement 1*), which send their axons to anatomically segregated mushroom body lobes, called the α, β, α′, β′, and γ lobes. We imaged APL in the vertical lobe (made of the α and α′ lobes) using GH146-QF driving QUAS-GCaMP3 (GH146-QF is specific to APL within the mushroom body lobes: *Pitman et al., 2011*; *Zhou et al., 2019*). When we activated only the α′β′ Kenyon cells using 853-GAL4 > dTRPA1 (*Silies et al., 2013*), remarkably, only the α′ lobe of APL responded (*Figure 3A–C,I*). The same occurred for the α lobe when we activated only αβ/γ Kenyon cells using mb247-GAL4 > dTRPA1 (*Figure 3D–F,I*) or only αβ Kenyon cells using c739-GAL4 > dTRPA1 or NP3061-GAL4 > dTRPA1 (*Figure 3G–I*). APL neurites in the unresponsive lobe failed to respond despite being only 2–3 μm away from responding neurites. This can most likely be explained by the fact that APL neurites in the vertical lobe mostly run in parallel along Kenyon cells and do not cross between the α and α′ lobes (*Takemura et al., 2017*). These results suggest that activity in APL fails to propagate over long distances.

To measure activity propagation within APL more systematically, we directly stimulated APL, to test whether local activation of APL in the mushroom body lobes (where Kenyon cell axons reside) would spread to the calyx (where Kenyon cell dendrites reside), and vice versa (*Figure 4*). To do this, we expressed GCaMP6f (*Chen et al., 2013*) and P2X2, an ATP-gated cation channel (*Lima and Miesenböck, 2005*), in APL, using a previously described intersectional driver for specifically labeling APL (NP2631-GAL4, GH146-FLP, tubP-FRT-GAL80-FRT [*Lin et al., 2014*]). *Drosophila* does not natively express P2X ATP receptors (*Littleton and Ganetzky, 2000*). We locally applied ATP to the tip of the vertical lobe or to the calyx by pressure ejection from a Picospritzer, while imaging both regions. This local stimulation evoked intense GCaMP6f responses at the site of ATP application, but not at the unstimulated site (*Figure 4A,D*), indicating that activity in APL attenuates to undetectable levels across the breadth of the neuron (about 250–300 μm; see below).

To rule out the possibility that all P2X2 driven excitation is always spatially restricted, we repeated the experiment but expressed P2X2 in Kenyon cells instead of APL, using mb247-LexA, which drives expression in all Kenyon cells (unlike mb247-GAL4) (*Pech et al., 2013*; *Pitman et al., 2011*). We still expressed GCaMP6f in APL, now using 474-GAL4 instead of the intersectional driver, to increase throughput (as this experiment did not require highly specific expression in APL). In contrast to stimulating APL, stimulating Kenyon cells in the calyx evoked a strong response in APL in both the calyx and the vertical lobe (*Figure 4B,E*), most likely because activating Kenyon cells in their dendrites drives them to spike, which then activates APL throughout the mushroom body. However, stimulating Kenyon cells at the tip of the vertical lobe activated APL locally, but not in the calyx (*Figure 4B, E*). Puffing ATP on negative control flies with no driver for P2X2 evoked no responses in APL (*Figure 4C,F*). Together, these findings show that activity within APL is spatially restricted.

## Quantification of activity spread in APL

Locally induced activity in APL declines to undetectable levels at the other end of the neuron, but how quickly does the activity decay along its journey? In particular, is activity in APL restricted enough for APL to signal differentially in different compartments of the MB lobes (i.e. the compartments defined by dopaminergic and output neuron innervation)? We examined this question by more systematically quantifying spread of activity throughout APL. We used volumetric imaging to record activity in the entire 3D extent of APL (~100 × 100 x 150 μm). We analyzed spatial activity in APL by creating a 3D 'backbone skeleton' of the mushroom body. This 'backbone' consisted of three lines, each one following the long axis of the vertical lobe, horizontal lobe, or peduncle/calyx. We used evenly spaced points every 20 μm along this backbone to divide the 3D volume of APL

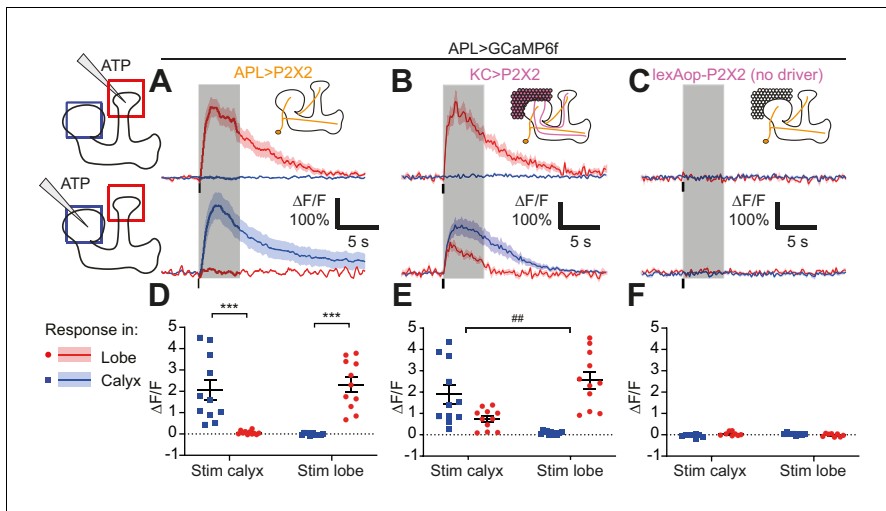

**Figure 4.** Activity in the APL neuron is spatially restricted. (**A–C**) Response of APL in the vertical lobe (red) or calyx (blue) to pressure ejection of 1.5 mM ATP (100 ms pulse, 12.5 psi) in the lobe or calyx, to stimulate APL (**A**), KCs (**B**), or negative control flies with lexAop-P2X2 but no LexA driver (**C**). Schematics show ROIs where responses were quantified (red/blue squares), and where ATP was applied (pipette symbol). Traces show mean ± SEM shading. Vertical lines on traces show time of ATP application; gray shading shows 5 s used to quantify response in (**D–F**). GCaMP6f and/or P2X2 expressed in APL via intersection of NP2631-GAL4 and GH146-FLP. P2X2 expressed in KCs via mb247-LexA. See **Supplementary file 1** for full genotypes. (**D–F**) Mean ΔF/F from (**A–C**), when stimulating APL (**D**), KCs (**E**), or negative controls (**F**). Error bars show SEM. n, given as # neurons (# flies): (**A,D**): 11 (9); (**B,E**): 11 (7); (**C,F**) 8 (4). ***p<0.001, t-tests with Holm Sidak multiple comparisons correction. ## p=0.004, paired t-test: the ratio of (response in unstimulated site)/(response in stimulated) site is higher for calyx stimulation than lobe stimulation. The online version of this article includes the following source data for figure 4:

**Source data 1.** Source data for **Figure 4D–F**.

into segments (Voronoi cells), and quantified fluorescence signals in APL in each segment-volume (**Figure 5A**, see Materials and methods). To compare the spatial extent of activity across flies, we normalized distances in these backbones to a 'standard' backbone based on the average across all flies (**Figure 5B**). Most neurites in APL run in parallel with this backbone (**Takemura et al., 2017**; **Wang et al., 2019**; **Scheffer et al., 2020**), making this backbone a meaningful axis along which to measure spread of APL activity.

To avoid the stochastic labeling of APL associated with the intersectional NP2631/GH146 driver, we developed another APL driver, VT43924-GAL4.2-SV40 (henceforth simply VT43924-GAL4.2), based on the original VT43924-GAL4 (**Wu et al., 2013**), as GAL4.2-SV40 typically gives ~2 x stronger expression than the original GAL4 (**Pfeiffer et al., 2010**). This line showed more reliable expression than the original VT43924-GAL4 line and labeled fewer neurons near the MB compared to 474-GAL4 (**Figure 5—figure supplement 1**), making it applicable for ATP stimulation of APL. We used VT43924-GAL4.2 for labeling APL in all subsequent experiments.

We locally activated APL by applying ATP to three zones – the tip of the vertical lobe, the calyx, and the horizontal lobe – and quantified the GCaMP6f response at evenly spaced intervals along the backbone skeleton. ATP stimulation of APL evoked a strong local response that gradually decayed as it propagated away from the stimulus site (**Figure 5**), with little change in the temporal dynamics of the response with distance (**Figure 5C–E**). Note that our temporal resolution (~5 Hz imaging rate) was not fast enough to reveal propagation of activity along APL neurites over time (which presumably occurs on the timescale of milliseconds). We co-ejected a red dye together with the ATP to reveal the spatial and temporal extent of stimulation as the ATP was removed by diffusion and perfusion. For example, the longer response from stimulation in the calyx reflects the slower disappearance of ATP (**Figure 5C–E**, lower panels).

Strikingly, activity evoked by stimulation at the tip of the vertical lobe decayed to an undetectable level by the branching point between the two lobes (**Figure 5G**). Stimulation in the other two zones elicited wider spread, but much of the activity spread actually arises from movement of ATP away

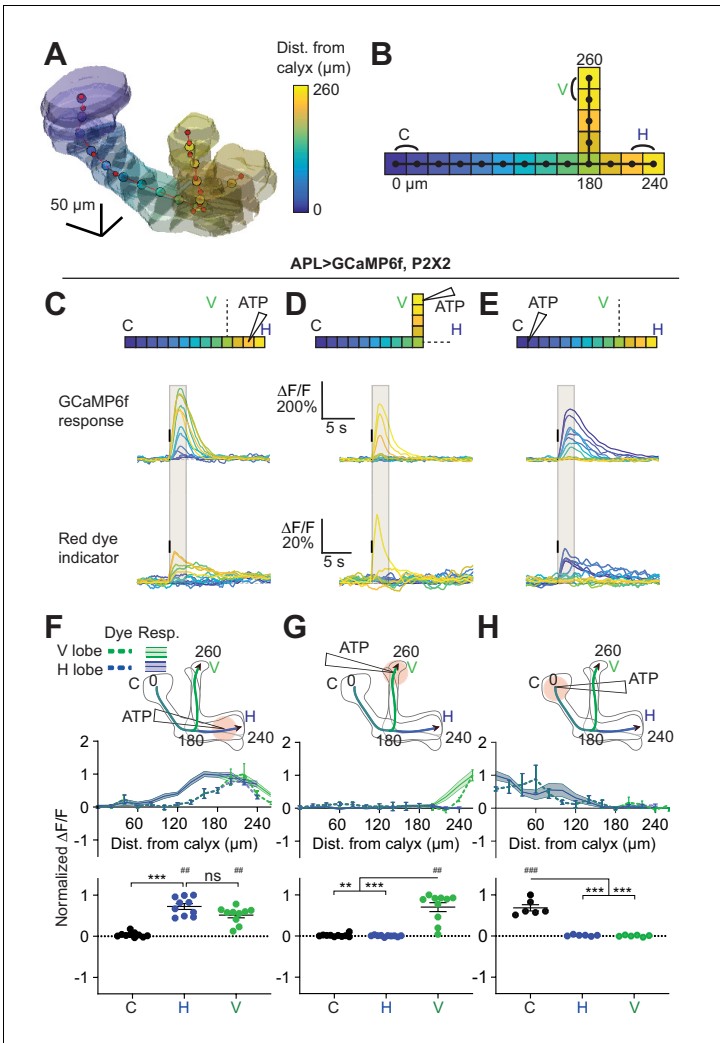

**Figure 5.** Quantification of activity spread in APL. (**A**) 3D visualization of an example mushroom body manually outlined using mb247-dsRed as an anatomical landmark. The backbone skeleton is shown as a red line passing through the center of each branch of the mushroom body, manually defined by the red dots. Evenly spaced nodes every 20 μm on the backbone subdivide the mushroom body into segments, here color-coded according to distance from the dorsal calyx (scale on right). (**B**) Schematic of the 'standard' backbone with distances (μm) measured from the dorsal calyx. Color code as in panel A. In this and later figures, signals are quantified using the two outermost segments of the calyx (C, black), vertical lobe (V, green), and horizontal lobe (H, blue), as shown here. (**C–E**) Traces show the time course of the response in each segment of APL, averaged across flies, when stimulating APL with 0.75 mM ATP (10 ms puff at 12.5 psi) in the horizontal lobe (**C**), vertical lobe (**D**), or calyx (**E**), in VT43924-GAL4.2>GCaMP6f,P2X2 flies. Upper panels: GCaMP6f signal. Lower panels: Red dye signal. The baseline fluorescence for the red dye signal comes from mb247-dsRed. Color-coded backbone indicates which segments have traces shown (dotted lines mean the data is omitted for clarity; the omitted data appear in panels **F–H**). Gray shading shows the time period used to quantify ΔF/F in (**F–H**). Vertical black lines indicate the timing of ATP application. (**F–H**) Mean response in each segment (ΔF/F averaged over time in the gray shaded period in **C–E**). The x-axis shows distance from the calyx (μm) along the backbone skeleton in the diagrams, and the color of the curves matches the vertical (green) and horizontal (blue) branches of the backbone. Solid lines with error shading show GCaMP responses; dotted lines with error bars show red dye. The responses in each panel were normalized to the highest responding segment (upper panels) or data point (lower panels). Error bars/shading show SEM. n, given as # neurons (# flies): (**C, F, D, G**) 10 (6), (**E, H**) 6 (4). ## p<0.01, ### p<0.001, one-sample Wilcoxon test or one-sample t-test, vs. null hypothesis (0) with Holm-Bonferroni correction for multiple comparisons. **p<0.01, ***p<0.001, Friedman test with Dunn's multiple comparisons test or repeated measures one-way ANOVA with Holm-Sidak's multiple comparisons test, comparing the stimulated vs the unstimulated sites. See **Supplementary file 2** for detailed statistics.

*Figure 5 continued on next page*

*Figure 5 continued*

The online version of this article includes the following figure supplement(s) for figure 5:

**Figure supplement 1.** Expression pattern of VT43924-GAL4.2.

from the ejection site, as revealed by spread of the co-ejected red dye, which spreads more widely with stimulation at the calyx and horizontal lobe than stimulation at the vertical lobe, most likely due to the physical structure of the brain tissue (dashed lines, *Figure 5*). These results confirm the spatially restricted activity shown in *Figure 4* and extend those results by systematically quantifying the extent of activity spread throughout APL.

## APL inhibits Kenyon cells mostly locally

Given that activity within APL is localized, we next asked whether APL's inhibitory effect on Kenyon cells is also localized. APL and Kenyon cells form reciprocal synapses throughout the entire extent of the mushroom body neuropil (Figure 8; *Takemura et al., 2017*; *Scheffer et al., 2020*; *Zheng et al., 2018*), suggesting that APL could theoretically inhibit Kenyon cells everywhere. However, it remains unknown if APL actually can inhibit Kenyon cells everywhere and if this inhibition remains local.

To test this, we again locally stimulated APL with ATP and used volumetric imaging to record activity not in APL but rather in Kenyon cells (expressing GCaMP6f under the control of mb247-LexA). Although Kenyon cells are mostly silent in the absence of odor, they do exhibit some spontaneous activity (*Turner et al., 2008*). Local stimulation of APL in the lobes strongly reduced the baseline GCaMP6f signal in Kenyon cells locally, but also had a smaller widespread effect. This can be seen in both color-coded time courses (*Figure 6A1,B1*) and plots of APL's inhibitory effect vs. distance along the backbone skeleton (*Figure 7A2,B2*). We saw qualitatively similar results when we

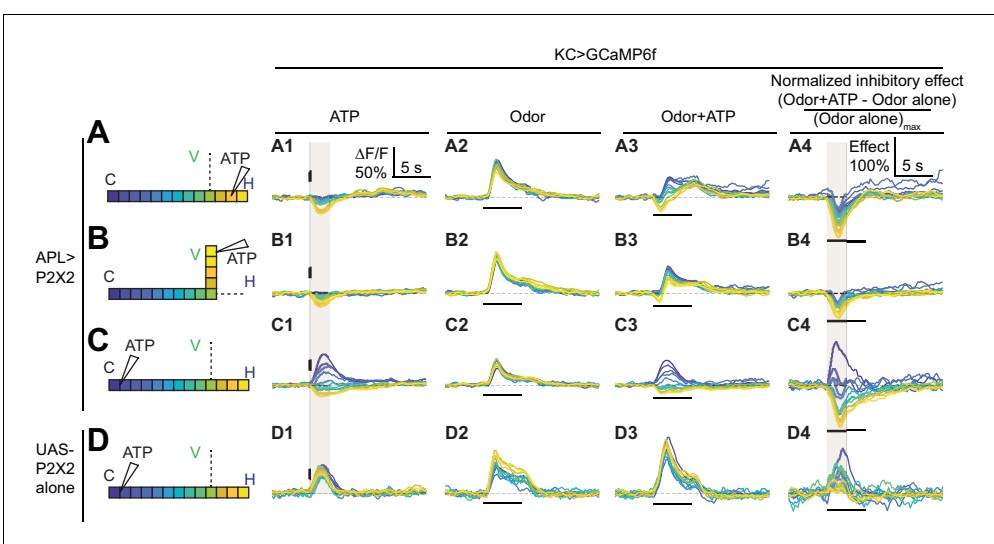

**Figure 6.** Time courses of APL's inhibitory effect on KC activity. Traces show the time course of the response in each segment of KCs, averaged across flies, when stimulating APL with 0.75 mM ATP (10 ms puff at 12.5 psi) in the horizontal lobe (**A**), vertical lobe (**B**), or calyx (**C**) in VT43924-GAL4.2>P2X2, mb247-LexA > GCaMP6f flies (no ATP stimulation in column 2). (**D**) shows responses in negative control flies (UAS-P2X2 alone). Columns 1–3 show responses of KCs to: (**A1–D1**) activation of the APL neuron by ATP, (**A2–D2**) the odor isoamyl acetate, (**A3–D3**) or both. (**A4–D4**) Normalized inhibitory effect of APL neuron activation on KC responses to isoamyl acetate (i.e. (column 3 - column 2)/(maximum of column 2)). The color-coded backbone indicates which segments have traces shown (dotted lines mean the data is omitted for clarity; the omitted data appear in *Figure 7*). Vertical and horizontal bars indicate the timing of ATP and odor stimulation, respectively. Gray shading in panels A1-D1 and A4-D4 indicate the intervals used for quantification in *Figure 7 A2–C2 and A4–C4*, respectively. The data in D1-D4 is quantified in *Figure 7—figure supplement 2*. n, given as # neurons (# flies): (**A1–A4, B1–B4**) 10 (9), (**C1–C4**) 9 (8), (**D1–D4**) 5 (3). Scale bar in **A1** applies to columns 1–3; scale bar in **A4** applies to column 4.

examined Kenyon cells' odor-evoked activity instead of spontaneous activity. Here, we measured the inhibitory effect as the difference between Kenyon cell odor responses with and without local stimulation of APL by ATP, normalized to the response without APL stimulation (*Figure 6A4,B4*, calculated from panels A2,B2 and A3,B3). Again, this normalized inhibitory effect was spatially non-uniform, declining in strength with distance away from the APL stimulation site (*Figure 7A4,B4*). Strikingly, when stimulating APL in the horizontal lobe, odor-evoked $Ca^{2+}$ influx in Kenyon cells was strongly inhibited in the lower vertical lobe, but less so in the tip of the vertical lobe (*Figure 7A2, A4*; difference between node 200 µm vs. 260 µm, p<0.01, paired t-test; see *Supplementary file 2*). For comparison, the same stimulation also induced stronger APL activity at node 200 µm than at node 260 µm (*Figure 7A1*). These results provide the first physiological evidence that local activity in APL can locally inhibit Kenyon cells. That inhibition of Kenyon cell axons can differ over such a short distance suggests that APL can differentially inhibit different mushroom body compartments defined by dopaminergic and output neuron innervation (see Discussion).

Surprisingly, puffing ATP on the calyx induced calcium influx in Kenyon cells in the calyx but inhibited calcium influx in the lobes (*Figures 6C* and *7C*). This apparently puzzling result can be explained by observing that puffing ATP on the calyx in negative control flies with UAS-P2X2 and no GAL4 also induced calcium influx in Kenyon cells, but did not inhibit it anywhere (*Figure 6D*, *Figure 7—figure supplement 2*). These results are best explained by leaky expression of P2X2 from the UAS-P2X2 transgene, either in Kenyon cells or projection neurons. In the negative control flies, the leaky P2X2 expression allows ATP to excite Kenyon cells in their dendrites, driving them to spike (hence ATP drives $Ca^{2+}$ influx throughout Kenyon cell dendrites and axons). However, in APL>P2X2 flies, the simultaneous activation of APL by ATP is strong enough to suppress Kenyon cell spikes despite the local excitation from leaky P2X2. The fact that there is still local $Ca^{2+}$ influx in the calyx despite local inhibition by APL can be explained by noting that both P2X2 channels and nicotinic acetylcholine receptors (nAChRs) are $Ca^{2+}$-permeable (*Lima and Miesenböck, 2005*; *Oertner et al., 2001*). Thus, we detect $Ca^{2+}$ entering directly through P2X2 or nAChRs in Kenyon cell dendrites (as opposed to through voltage-gated $Ca^{2+}$ channels), but inhibition from APL shunts away the resulting depolarization and stops the Kenyon cells from spiking, explaining how local APL stimulation at Kenyon cell dendrites blocks both spontaneous and odor-evoked activity in Kenyon cell axons. These results show that sufficiently strong local inhibition in Kenyon cell dendrites can silence Kenyon cells throughout their axons.

The inhibitory effect was sometimes less localized than APL activity itself. For example, when ATP was ejected at the tip of the vertical lobe, APL activity decreased to zero by 100 µm from the ejection site and there was no activity in the horizontal lobe (*Figures 5D,G* and *7B1*), whereas the inhibitory effect on Kenyon cells persisted into the peduncle and the horizontal lobe (*Figure 7B2*), even into the calyx (>200 µm away) in the case of the inhibitory effect on Kenyon cell odor-evoked responses (*Figure 7B4*). However, this more extended inhibitory effect on the calyx was significantly smaller than the local inhibitory effect on the vertical lobe. These results suggest that APL can, to a modest extent, inhibit areas of Kenyon cells beyond areas where APL itself is active (see Discussion).

## Local GABA application mimics the inhibitory effect of local APL stimulation

How could APL inhibit Kenyon cells in areas where it itself is not active? One explanation could be that because we are measuring GCaMP6f signals and not voltage, APL might be depolarized even where GCaMP6f signal is not increased. However, synaptic release still presumably requires $Ca^{2+}$ influx, so even if APL is depolarized without $Ca^{2+}$ influx, it should not release GABA where there is no $Ca^{2+}$ influx. Still, GCaMP6f might not be sensitive enough to detect very low levels of $Ca^{2+}$ influx. Therefore, to test whether APL releases GABA over a wider area than its area of GCaMP6f increase, we replaced APL stimulation with direct application of GABA (here, we drove GCaMP6f expression using OK107-GAL4, which drives expression in all Kenyon cells like mb247-LexA). We reasoned that if local ATP stimulation causes APL to release GABA over a wider region than APL's own GCaMP6f signal, then directly applying GABA should cause a more spatially restricted inhibitory effect than applying ATP (which would evoke wider release of GABA from APL).

In fact, directly applying GABA caused the same inhibitory effect on Kenyon cells as locally stimulating APL with ATP: strongest at the site of stimulation but still affecting them far from the site of stimulation, to a lesser extent (*Figure 7*, columns 3 and 5; traces in *Figure 7—figure supplement*

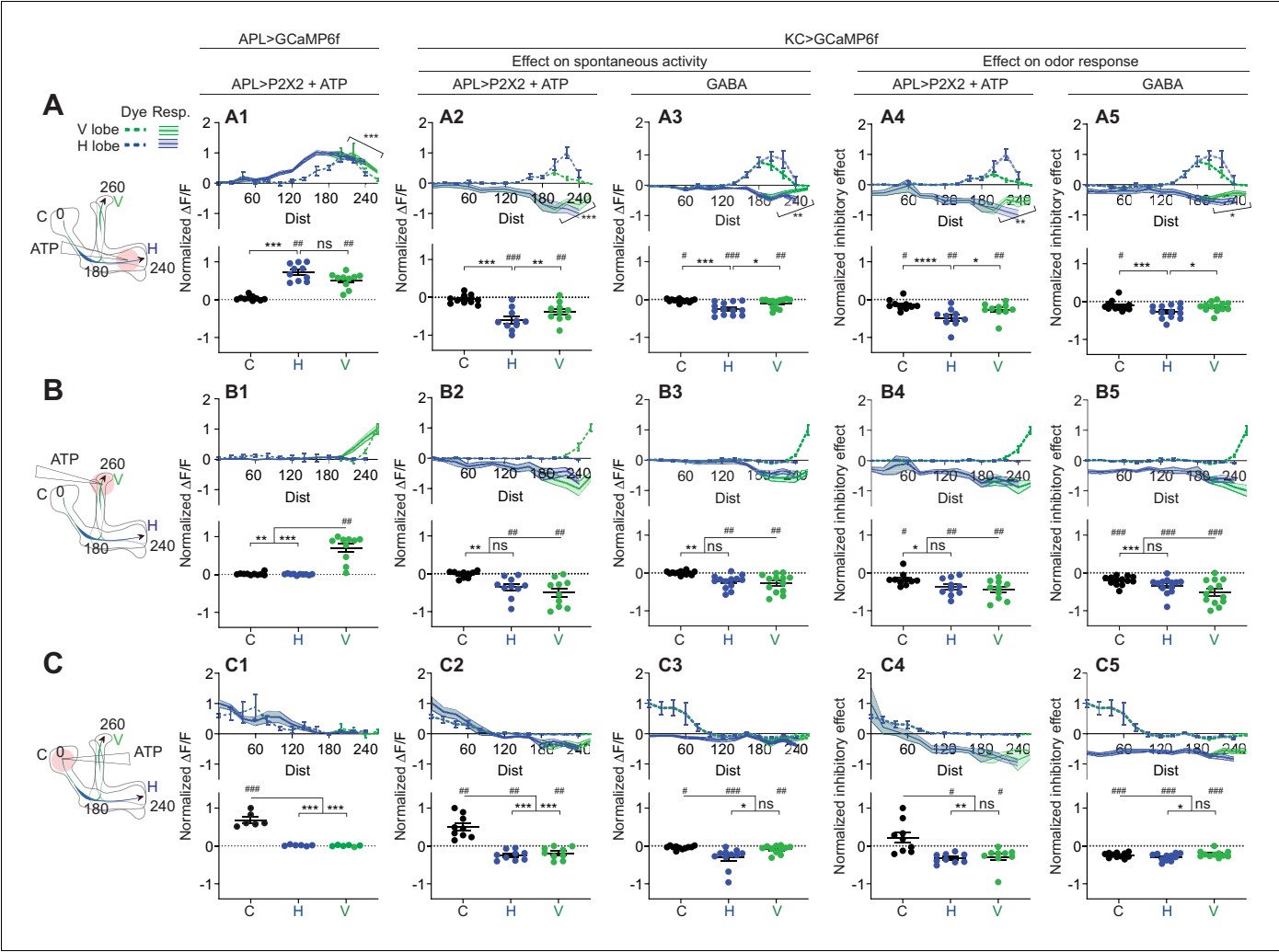

**Figure 7.** Quantification of the inhibitory effect of GABA or the APL neuron on KC activity. Rows: Local application of ATP (0.75 mM) or GABA (7.5 mM) in the horizontal lobe (**A1–A5**), vertical lobe (**B1–B5**) or calyx (**C1–C5**). Columns: Column 1: APL's response to ATP stimulation (**A1–C1**) in VT43924-GAL4.2>GCaMP6f,P2X2 flies, repeated from *Figure 5* for comparison. Columns 2–3: KC responses to local activation of APL by ATP (**A2–C2**) or to GABA application (**A3–C3**). Columns 4–5: Normalized inhibitory effect of APL activation (**A4–C4**) or GABA application (**A5–C5**) on KC responses to isoamyl acetate. Genotypes: for ATP (columns 2,4): VT43924-GAL4.2>P2X2, mb247-LexA > GCaMP6f; for GABA (columns 3,5): OK107-GAL4 > GCaMP6f. Data shown are mean responses in each segment (averaged over time in the gray shaded periods in *Figure 6*). The x-axis ('Dist') shows distance from the calyx (µm) along the backbone skeleton in the diagrams (left), and the color of the curves matches the vertical (green) and horizontal (blue) branches of the backbone. Solid lines with error shading show GCaMP responses; dotted lines with error bars show red dye. The responses were normalized to the segment (upper panels) or data point (lower panels) with the largest absolute value across matching conditions (columns 2+3, or columns 4+5). The baseline fluorescence for the red dye comes from bleedthrough from the green channel; only trials without odour were used for red dye quantification, in these trials, the change in green bleedthrough (~10–40%) is negligible compared to the increase in red signal (150–300%). Error bars/shading show SEM. n, given as # neurons (# flies): (**A1, B1**) 10 (6), (**C1**) 6 (4), (**A2, A4, B2, B4**) 10 (9), (**C2, C4**) 9 (8) (**A3, A5, B3, B5**) 13 (8), (**C3, C5**) 11 (6). # p<0.05 ### p<0.001, one-sample Wilcoxon test, or one-sample t-test, vs. null hypothesis (0) with Holm-Bonferroni correction for multiple comparisons. *p<0.05, ***p<0.001, Friedman test with Dunn's multiple comparisons test, or repeated-measures one-way ANOVA with Holm-Sidak multiple comparisons test, comparing the stimulated site vs. the unstimulated sites. Diagonal brackets in (**A1–A5**), paired t-test or Wilcoxon test comparing the response at segment 200 vs. 260 on the vertical branch. See *Supplementary file 2* for detailed statistics.

The online version of this article includes the following source data and figure supplement(s) for figure 7:

**Source data 1.** Source data for *Figure 7* and *Figure 7—figure supplement 2*.

**Figure supplement 1.** Time courses of GABA's effect on KC activity.

**Figure supplement 2.** Quantification of effect of ATP on negative control (UAS-P2X2 only).

*1*). In particular, GABA applied to the tip of the vertical lobe inhibited Kenyon cells throughout the horizontal lobes. In the case of odor-evoked activity, this inhibition extended to the peduncle and calyx, although significantly weaker than in the vertical lobe (*Figure 7B3,B5*, compare to *Figure 7B2,B4*). These results argue against the possibility that APL releases GABA more widely than its area of activity and suggest that APL's inhibitory effect extends beyond the zone of its own activity, although weaker than where it is active (see Discussion).

In other respects as well, puffing GABA elicited the same effects above noted for locally activating APL with ATP. Locally inhibiting Kenyon cells in the calyx inhibited them throughout the lobes (*Figure 7C3,C5*, compare to *Figure 7C2,C4*, now without the confounding effects of leaky P2X2 expression). With local inhibition in the horizontal lobe, Kenyon cells were more inhibited in the lower vertical lobe than at the vertical lobe tip (*Figure 7A3,A5*, compare to *Figure 7A2,A4*). The close correspondence of effects of GABA and ATP application indicates that the observed localization of ATP-induced $Ca^{2+}$ influx in APL accurately reflects the localization of GABA release.

## Connectomic analysis predicts that each Kenyon cell disproportionately inhibits itself

We next asked how local inhibition by APL affects lateral inhibition between Kenyon cells. APL is thought to act through all-to-all feedback inhibition to sparsen and decorrelate population odor responses in Kenyon cells (*Lin et al., 2014*). Such a function assumes that each Kenyon cell inhibits all Kenyon cells via APL roughly equally and indeed, decorrelation should theoretically work better if all Kenyon cells inhibit each other than if each Kenyon cell inhibits only itself (see Appendix 1). However, if a Kenyon cell activates APL locally and APL is not evenly activated throughout, it may inhibit other Kenyon cells unevenly, that is, some more than others. Such uneven lateral inhibition would depend both on how activity spreads within APL and on how APL-KC and KC-APL synapses are spatially arranged on APL neurites. In particular, if activity in APL decays with distance, activating Kenyon cell 1 (KC1) would inhibit another Kenyon cell (KC2) more strongly if KC1-APL synapses are close to APL-KC2 synapses (as measured along APL's neurites) than if they are far apart. Thus, the degree to which KC1 inhibits KC2 can be described by the weighted count of every pair of KC1-APL and APL-KC2 synapses (see Materials and methods), where the weight of each pair should fall off exponentially with the distance between the two synapses, given that cable theory predicts that signals decay as exp(-x/space constant) (*Hodgkin and Rushton, 1946*; *Figure 8G*).

To test whether KC-APL synapses are spatially arranged to allow uneven lateral inhibition, we analyzed the hemibrain connectome dataset released by Janelia FlyEM and Google (v. 1.1) (*Scheffer et al., 2020*), which contains a fully segmented APL and annotated KC-APL and APL-KC synapses. All 1927 traced Kenyon cells in this volume form reciprocal synapses with APL (49.6 ± 17.9 APL-KC and 52.6 ± 13.4 KC-APL synapses per KC; mean ± s.d.) in both their dendrites and axons (example Kenyon cells shown in *Figure 8A*, *Figure 8—figure supplement 1*). APL-KC synapses are polyadic (a single presynaptic density on APL contacts multiple KCs). We mapped all annotated APL-KC and KC-APL synapses onto APL's neurite skeleton (182,631 connected nodes, total length 80.2 mm; 95,678 and 101,430 unique synapses mapped to 15,656 and 96,867 unique locations on APL's neurite skeleton for APL-KC and KC-APL synapses, respectively) (*Figure 8B,C*) and measured the distance from every APL-KC to every KC-APL synapse along APL's neurite skeleton (i.e. the detailed morphology from the connectome, not the backbone skeleton of *Figures 5–7*).

To determine what space constant to use in our analysis, we mapped our recordings onto the reconstructed APL by aligning the connectome neurite skeleton to the average backbone skeleton in *Figure 5B*, and dividing it into segments (Voronoi cells) along the backbone as in *Figures 5–7*. Here, we used segments spaced 10 μm apart to increase spatial resolution (in *Figures 5–7* we used 20 μm spacing to reduce variability across individual flies). We took the red dye signal in each segment from *Figure 5*, averaged across flies, at the time of the peak signal in the segment receiving the most red dye. Because these signals were weak and noisy (peak ΔF/F ~ 20–30%, compared to 300% for GCaMP6f), we fitted exponential decay curves to approximate the ATP stimulus that each segment received. We then simulated *Figure 5* by placing ~10,000 randomly distributed points on the APL neurite skeleton (*Figure 8D*) and 'stimulating' each segment of APL according to our curve fits of red dye signal (*Figure 8E*). Given this localized stimulus, we calculated how much signal an average point in each segment of APL would receive from all other points in APL, given a particular normalized space constant. 'Normalized space constant' means the space constant taking into

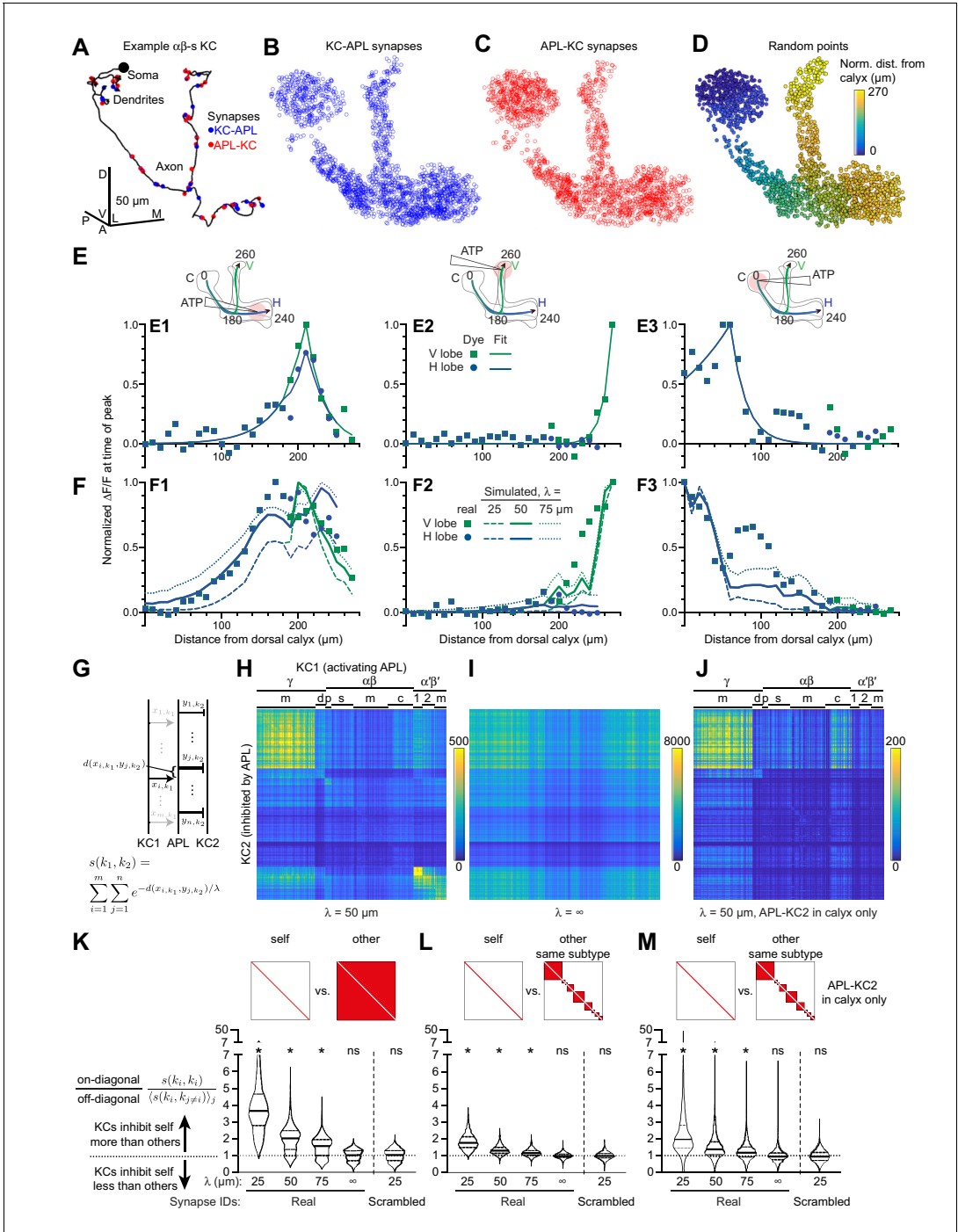

**Figure 8.** Localized activity and KC-APL anatomy suggest that KCs inhibit themselves more than other KCs. (A) Line drawing of an example αβ$_s$ KC (body ID 5813078552). Soma shown by black circle (not to scale). KC-APL (blue circles) and APL-KC (red circles) synapses appear in the KC's dendrites and axon. Dendritic KC-APL synapses are consistent with reports of presynaptic release from Kenyon cells in the calyx (*Christiansen et al., 2011*). Scale bar for (A–D), 50 μm, D, dorsal, V, ventral, A, anterior, P, posterior, L, lateral, M, medial. (B,C) 2000 randomly selected synapses from the set of 96,867 unique KC-APL synapse locations (B) and 15,656 unique APL-KC synapse locations (C). The relative lack of KC-APL and APL-KC synapses in the posterior peduncle is consistent with the lack of syt-GFP signal in both APL and Kenyon cells in this zone, thought to be the Kenyon cells' axon initial segment (*Trunova et al., 2011*; *Wu et al., 2013*). (D) 2000 random points along APL's neurite skeleton, color-coded by normalized distance from the dorsal calyx along the average backbone skeleton in *Figure 5*. (E) Red dye signal from *Figure 5* in 10 μm segments (dots), with fitted exponential decay curves (lines), for stimulation in the horizontal lobe (E1), vertical lobe (E2), and calyx (E3). The slight discontinuity in the curve fit for (E1) arises because the calyx-to-junction branch has slightly different fits for the calyx-to-vertical vs. calyx-to-horizontal branches; where the two merge, we averaged the fits together. (F) Real activity in APL from *Figure 5* (dots) compared to simulated activity for different space constants (lines), given

*Figure 8 continued on next page*

*Figure 8 continued*

localized stimulation defined by fitted curves in (**E**) in the horizontal lobe (**F1**), vertical lobe (**F2**), and calyx (**F3**). (**G**) Schematic of metric s(k1,k2) for predicting how strongly KC1 inhibits KC2 via APL given the space constant λ and the relative distances (d) between KC1-APL ($x_{i,k1}$) and APL-KC2 ($y_{j,k2}$) synapses along APL's neurite skeleton. The variable thickness of the APL-KC2 synapses depicts how synapses closer to the KC1-APL synapse ($x_{i,k1}$) are weighted more strongly. The other KC1-APL synapses are greyed out to show that although they are part of the double summation, the focus in the diagram is on distances between $x_{i,k1}$ and APL-KC2 synapses. (**H**) s(k1,k2) for all pairs of KCs, for space constant 50 μm. αβ, α′β′, γ subtypes as annotated in the hemibrain v1.1: γ subtypes: m, main; d, dorsal; γ-t subtype not labeled (t for thermo) – contains 8 KCs placed on the grid between (γ-m and γ-d). αβ: p, posterior; s, surface; c, core; m, intermediate KCs between surface and core. α′β′: 1, ap1; 2, ap2; m, medial, where ap corresponds to PAM-β′1ap and m corresponds to PAM-β′1 m (personal communication, S. Takemura). Note that s(k1,k2) is higher along the diagonal. KCs were sorted by subtype as annotated in the hemibrain dataset, and reordered by hierarchical clustering to place similar KCs within the same subtype next to each other. (**I**) As (**H**) except space constant is infinite. (**J**) As (**H**) except only including APL-KC synapses ($y_{j,k2}$ in panel **F**) that are in the calyx. In (**I–J**), the order of KCs is preserved from (**H**). (**K–M**) The violin plots show, for each KC1, the ratio of s(k1,k1) (self-inhibition) vs. the average of s(k1,k2) across all k2 (inhibiting other KCs), given different space constants (λ), where k2 includes all other KCs (**K**), or all other KCs within the same subtype (**L**). Referring to (**H–J**), this ratio is each on-diagonal pixel divided by the average of the off-diagonal pixels in corresponding column. (**M**) As (**L**) except only including APL-KC synapses in the calyx. The squares above the violin plots mark in red which pixels are being analyzed from a heat map of s(k1,k2) as in panel **H**: that is, for 'self', the diagonal pixels; for 'other', the off-diagonal pixels; for 'other, same subtype', the off-diagonal pixels for KCs of the same subtype. 'Synapse IDs': For 'Scrambled' (but not 'Real'), the identities of which Kenyon cell each KC-APL or APL-KC synapse belonged to were shuffled. Solid horizontal lines show the median; dotted lines show 25% and 75% percentile. n = 1923 KCs (4 of 1927 KCs annotated as KCγ-s1 through s4 – 's' for 'special' – are excluded as there is only one example of each). *p<0.0001, different from 1.0 by Wilcoxon test with Holm-Bonferroni correction for multiple comparisons.

The online version of this article includes the following source data and figure supplement(s) for figure 8:

**Source data 1.** Source data for *Figure 8*.
**Figure supplement 1.** Example Kenyon cells.
**Figure supplement 2.** Additional data for *Figure 8*.
**Figure supplement 2—source data 1.** Source data for *Figure 8—figure supplement 2*.
**Figure supplement 3.** Self- vs. other-inhibition in different Kenyon cell subtypes.
**Figure supplement 3—source data 1.** Source data for *Figure 8—figure supplement 3*.
**Figure supplement 4.** Diameter of APL neurites.

account the varying diameter of APL neurites; we modeled space constants as varying with the square root of the neurite radius (see Materials and methods). Note that for simplicity, we did not consider branching effects in APL.

Given that the space constant $\lambda = \sqrt{\frac{rR_m}{2R_a}}$ (r = radius, $R_m$ = specific membrane resistance, $R_a$ = specific axial resistance), we first tested space constants derived from reported estimates of the ratio $\frac{R_m}{R_a}$ in *Drosophila* neurons (*Scheffer et al., 2020*). At the low end, we tested $\frac{R_m}{R_a}$ = 0.0907 m, derived from HS cells of the lobula plate tangential complex, which are similar to APL in being non-spiking neurons with enormous neurite arbors ($R_m$ = 0.8166 Ωm², $R_a$ = 9 Ωm) (*Cuntz et al., 2013*). Note that although (*Cuntz et al., 2013*) use $R_a$ = 4 Ωm in their model as $R_a$ = 9 Ωm is considered too high compared to other invertebrate neurons (*Borst and Haag, 1996*), they note that $R_a$ = 9 Ωm is a better fit to their data. This ratio corresponds to a normalized space constant of λ = 95 μm (see Materials and methods). At the high end, we tested $\frac{R_m}{R_a}$ = 2.0 m, derived from an olfactory projection neuron ($R_m$ = 2.04 Ωm², $R_a$ = 1.02 Ωm) (*Gouwens and Wilson, 2009*), corresponding to λ = 448 μm.

Surprisingly, with both of these values, activity spread too far in APL, to areas where *Figure 5* found no activity (*Figure 8—figure supplement 2B*), suggesting that APL's normalized space constant is shorter than 95 μm and that $\frac{R_m}{R_a}$ might be unusually low in APL (see Discussion). Next, we tested shorter space constants: 25, 50, or 75 μm. Overall, 50 μm gave the best fit: 25 μm did not spread activity enough, while 75 μm spread activity too far (*Figure 8F*). Similar results were obtained when treating APL as having the same neurite radius throughout, or when modeling activity as decaying along the backbone skeleton rather than the connectome neurite skeleton (*Figure 8—figure supplement 2C,D*). The latter showed some differences, likely reflecting areas where APL's neurites do not simply travel parallel to the backbone.

Using these space constants, our model predicts that Kenyon cells disproportionately inhibit some Kenyon cells more than others (*Figure 8G–M*). In particular, each Kenyon cell inhibits itself more strongly than it inhibits other individual Kenyon cells on average (*Figure 8K–M*). (This can be

seen in the brighter pixels along the diagonal in *Figure 8H,J*.) This is due partly to the fact that γ, αβ$_p$ and α′β′ Kenyon cells inhibit other Kenyon cells of the same subtype more than Kenyon cells of other subtypes. (This can be seen as the large blocks along the diagonal in *Figure 8H*). However, even when comparing self-inhibition to inhibition of only other Kenyon cells of the same subtype, each Kenyon cell inhibits itself more than it inhibits other individual Kenyon cells on average (*Figure 8L*).

These results arise from treating every KC-APL and APL-KC synapse equally. In reality, Kenyon cell spiking is driven by dendritic input in the calyx, as confirmed by our finding that activity in Kenyon cells can be suppressed throughout the mushroom body by locally activating APL or applying GABA in the calyx (*Figures 6* and *7*). Therefore, we repeated this analysis, taking into account only APL-KC synapses in the calyx. Again, our model predicts that each Kenyon cell inhibits itself more than it inhibits other individual Kenyon cells on average (*Figure 8J,M*). This imbalance is consistent across different Kenyon cell subtypes (except αβ-p Kenyon cells, likely because they form few synapses with APL in the calyx; they still show more self- than other-inhibition when considering all APL-KC synapses; *Figure 8—figure supplement 3*). Of course, despite this imbalance, given that there are ~2000 Kenyon cells, the sum total of lateral inhibition onto a Kenyon cell would still be stronger than its own self-inhibition (see Discussion).

Similar results were obtained when treating APL as having the same neurite radius throughout (*Figure 8—figure supplement 2E*). Importantly, this imbalance decreases with longer space constants (although it remains non-zero even at the space constant corresponding to the $\frac{R_m}{R_a}$ ratio of HS cells; *Figure 8—figure supplement 2F*), and disappears entirely if the space constant is infinite (at λ = 50 μm, the median imbalance is ~40%). Moreover, the imbalance arises from the particular spatial relations between synapses of different Kenyon cells, because the imbalance disappeared when we shuffled the identities of which Kenyon cell each KC-APL or APL-KC synapse belonged to (*Figure 8K–M*). Thus, our physiological measurements of localized activity of APL, combined with the spatial arrangements of KC-APL and APL-KC synapses, predict that Kenyon cells disproportionately inhibit themselves compared to other individual Kenyon cells.

## Discussion

We have shown that activity in APL is spatially restricted intracellularly for both sensory-evoked and local artificial stimulation. This local activity in APL translates into a spatially non-uniform inhibitory effect on Kenyon cells that is strongest locally and becomes weaker farther from the site of APL stimulation. Finally, combining physiological and anatomical data – APL's estimated space constant and the spatial arrangement of KC-APL and APL-KC synapses – predicts that each Kenyon cell disproportionately inhibits itself more than other individual Kenyon cells.

The locust equivalent of APL is called GGN ('giant GABAergic neuron') (*Papadopoulou et al., 2011*). A compartmental model of GGN predicts that local current injection in the α lobe should lead to depolarization throughout GGN (at least 500 μm across) though the depolarization should decrease with distance from the injection site (*Ray et al., 2020*). In contrast, we found that local stimulation of APL with P2X2 + ATP decayed to undetectable levels within as little as 100 μm. Of course, undetectable ΔF/F in GCaMP6f does not necessarily mean zero depolarization or synaptic release: GCaMP6f may not be sensitive enough to detect very small amounts of calcium influx, and as little as 2 mV depolarization can modulate neurotransmitter release in non-spiking insect interneurons (*Burrows and Siegler, 1978*). Still, local stimulation of APL with P2X2 + ATP inhibits Kenyon cell odor responses with a similar spatial extent as local application of GABA, implying that GCaMP6f signals accurately predict GABA release (see below). Thus, while keeping in mind the different methods used for GGN vs. APL, activity seems to spread less far in APL than in GGN.

What may explain this apparent difference? The GGN model used specific membrane and axial resistances with an $\frac{R_m}{R_a}$ ratio (0.7 to 10 m) higher than the value that fit our experimental measurements of activity spread in APL to APL's detailed anatomy ($\frac{R_m}{R_a}$ = 0.025 m for λ = 50 μm). This difference could easily explain the different results (lower $\frac{R_m}{R_a}$ means shorter space constant, thus less activity spread), but as $R_m$ and $R_a$ have not been measured in either APL or GGN, we cannot say which values are correct. Although our best-fit value of $\frac{R_m}{R_a}$ is very low in the range of published estimates (0.02 to >6 m) (*Borst and Haag, 1996*), other *Drosophila* neurons also show unusual electrical

features — Kenyon cells have input resistance >10 GΩ (*Turner et al., 2008*), and the best fit to data for HS cells suggest a specific axial resistance of 9 Ωm (*Cuntz et al., 2013*) — and APL is already morphologically unusual, so an unusually low $\frac{R_m}{R_a}$ ratio may not be surprising. Moreover, active conductances (e.g. voltage-gated K$^+$ channels) might decrease $R_m$ when APL is activated, so that $\frac{R_m}{R_a}$ is actually higher than 0.025 m at resting potential.

In addition, although GGN and APL are both widefield inhibitory interneurons in the mushroom body, they have striking morphological differences. For example, GGN separately innervates the lobes and calyx with numerous thin, branching processes; these tangled innervations are joined by relatively thick processes (almost 20 μm in diameter) (*Ray et al., 2020*). In contrast, APL innervates the lobes continuously with numerous neurites that emerge in parallel from the calyx (*Mayseless et al., 2018*; *Scheffer et al., 2020*). In the hemibrain connectome (*Scheffer et al., 2020*), these neurites average ~0.5 μm in diameter throughout APL; ~95% of neurite length is less than 1 μm in diameter and the maximum diameter is ~3 μm (*Figure 8—figure supplement 4*). The narrowness of these neurites alone would predict that activity propagates poorly in APL compared to GGN. GGN's morphology actually more resembles *Drosophila*'s larval APL, which has separate innervations of the lobes and calyx joined by a single process (*Eichler et al., 2017*; *Masuda-Nakagawa et al., 2014*). The larval APL has pre-synapses only in the calyx, while adult APL has pre-synapses everywhere (*Figure 8A*). It is tempting to speculate that the locust and larval *Drosophila* mushroom bodies use global activity in APL/GGN to send feedback from Kenyon cell axons to Kenyon cell dendrites, whereas the adult *Drosophila* mushroom body uses local activity in APL to locally inhibit Kenyon cells everywhere. The larval APL might be small enough for activity to spread passively from lobes to calyx (~100 μm) or might have different ion channel composition from adult APL.

We measured local activity and local effects of APL by locally puffing ATP or GABA on the mushroom body. This technique requires two caveats but we argue that neither substantially affects our conclusion that APL acts mostly locally. First, how spatially precise was ATP/GABA application? The co-ejected red dye indicator spread tens of microns from the ejection site (decay to half-strength ~10–25 μm; see *Supplementary file 2*). This spatial resolution is close to that of other techniques for local in vivo stimulation like two-photon optogenetic stimulation (decay to half-strength 12 μm horizontal, 24 μm axially [*Packer et al., 2015*]). How precisely does the red dye indicate the spread of the bioactive compound (ATP/GABA)? This is difficult to answer with certainty, as it is unclear how much of the spread of dye/ATP/GABA arises from bulk flow from ejection from the pipette (which should affect dye, ATP and GABA equally) vs. diffusion (which depends on molecular weight and molecular volume). To the extent that ATP or GABA spread differently from the red dye, they are likely to spread farther, as they have lower molecular weights (dye: 1461; ATP: 507; GABA: 103), meaning that our stimulation is less spatially restricted than the dye suggests. Thus, our results likely overestimate, rather than underestimate, how far APL activity and APL's inhibitory effect spread from the stimulation site. If the space constant is indeed lower than our normalized estimate of 50 μm, then the imbalance between Kenyon cells inhibiting themselves vs. others is likely even more severe (compare λ = 25 μm and 50 μm on *Figure 8K–M*). In any case, in comparing the spatial spread of activity in APL vs. APL's inhibitory effect, we used the same protocol for ATP stimulation, so these results are directly comparable even if the exact distance of ATP spread is uncertain.

Second, ATP application might activate non-APL neurons that have leaky expression of P2X2 from the UAS-P2X2 transgene. We accounted for leaky P2X2 expression by testing UAS-P2X2 only flies when imaging Kenyon cells, but not when imaging APL (as the same GAL4 driver controlled both GCaMP6f and P2X2). Neurons activated by leaky P2X2 expression could excite or inhibit APL (directly or indirectly). If they excite APL (as with Kenyon cells, which are indeed activated by ATP puffed on the calyx in UAS-P2X2 only flies), the ectopic activation should make us overestimate, rather than underestimate, the spread of APL activity (as in the previous paragraph). On the other hand, if ectopically activated neurons inhibit APL, restricted activity spread in APL might be explained by inhibition rather than APL's intrinsic properties. However, if this scenario were true, puffing ATP on UAS-P2X2 only flies would inhibit APL and thereby increase the activity of Kenyon cells. Contrary to this prediction, when we puffed ATP on the horizontal or vertical lobe of KC>GCaMP6f, UAS-P2X2 control flies, Kenyon cell activity was unaffected (*Figure 7—figure supplement 2*; presumably any leaky expression of P2X2 in Kenyon cells was too low for ATP applied to

their axons to excite them). Therefore, it is unlikely that leaky expression of P2X2 in inhibitory neurons affects APL activity enough to make us underestimate activity spread in APL.

We showed that APL can locally inhibit Kenyon cells in multiple locations in the mushroom body. Remarkably, inhibition of $Ca^{2+}$ influx in Kenyon cell axons could be stronger closer to the axon initial segment (in the posterior peduncle [*Trunova et al., 2011*]) than farther from it, for example, stronger at the lower vertical lobe than the tip of the vertical lobe (*Figure 7A*), both when stimulating APL with ATP or when directly applying GABA. At first glance, this observation is puzzling given the direction of action potential travel. GABA should act on Kenyon cells by suppressing depolarization through shunting inhibition in both dendrites and axons, as the $GABA_A$ receptor Rdl is expressed in both (*Liu et al., 2007*). Therefore, our observation of stronger proximal than distal inhibition appears to suggest that as action potentials travel from the axon initial segment toward the distal tip of the axon, they can enter a zone of shunting inhibition and be suppressed, yet recover on the other side. Such a scenario could occur if depolarization is suppressed enough to reduce voltage-gated $Ca^{2+}$ influx in a local zone, yet remains sufficient on the other side of the inhibitory zone to trigger enough voltage-gated $Na^+$ channels to regenerate the action potential. However, this interpretation seems unlikely. More likely, local inhibition might particularly suppress $Ca^{2+}$ influx rather than depolarization, perhaps by acting through $GABA_B$ receptors; Kenyon cells express $GABA_B$R1 and $GABA_B$R2 (*Aso et al., 2019*; *Crocker et al., 2016*; *Croset et al., 2018*; *Davie et al., 2018*) and APL inhibits KCs partly via $GABA_B$ receptors (*Inada et al., 2017*), although their subcellular localization is unknown. Given that synaptic vesicle release requires $Ca^{2+}$ influx, such inhibition would still locally suppress Kenyon cell synaptic output.

Local inhibition of Kenyon cell output predicts that activity of MBONs near the site of APL activation would be more strongly inhibited than MBONs far away. This prediction may be tested in future experiments, for example locally stimulating APL in the tip of the vertical lobe and comparing the inhibitory effect on MBONs in nearby compartments like α3 and α′3 vs. on MBONs in distant compartments like γ5 and β2.

While APL inhibits Kenyon cells locally, the inhibitory effect spreads somewhat more widely than APL's own activity, though weakly (*Figure 7A4-5,B4-5*). Local GABA application produces similar results to locally activating APL with ATP (*Figure 7*), suggesting that GCaMP6f signals in APL accurately predict GABA release. How can APL inhibit Kenyon cells where it itself is not active? Wider inhibition when inhibiting Kenyon cell dendrites (*Figure 7C*) can be easily explained as blocking action potentials, but the wider inhibition when inhibiting the axons is more puzzling. We speculate that this result might arise from wider network activity. For example, Kenyon cells form recurrent connections with DPM and dopaminergic neurons and form synapses and gap junctions on each other (*Liu et al., 2016*; *Takemura et al., 2017*). Through such connections, an odor-activated Kenyon cell might excite a neighboring Kenyon cell's axon; the neighbor might passively spread activity both forward and backward or, not having fired and thus not being in a refractory period, it might even be excited enough to propagate an action potential back to the calyx. Alternatively, Kenyon cells indirectly excite antennal lobe neurons in a positive feedback loop (*Hu et al., 2010*). In these scenarios, locally puffing GABA or activating APL artificially in the vertical lobe tip would block the wider network activity, thus reducing $Ca^{2+}$ influx in the calyx. Although the calyx was not activated when stimulating Kenyon cells in the vertical lobe tip in *Figure 4B*, this might be because simultaneous activation of all Kenyon cells (but not odor-evoked activation of ~10% of Kenyon cells) drives strong-enough local APL activation to block wider network activity. Future experiments may test these possibilities.

By combining our physiological measurements of localized activity with the detailed anatomy of the hemibrain connectome (*Scheffer et al., 2020*), we built a model that predicts that the average Kenyon cell inhibits itself more than it inhibits other individual Kenyon cells. This prediction is supported by previous experimental results that some Kenyon cells can inhibit some Kenyon cells more than others, and that an individual Kenyon cell can inhibit itself (*Inada et al., 2017*). Our model goes beyond these results in predicting that the average Kenyon cell actually *preferentially* inhibits itself (*Figure 8*), and by explaining how differential inhibition can arise from local activity in APL and the spatial arrangement of KC-APL and APL-KC synapses.

Note that these results do not contradict our previous findings that Kenyon cell lateral inhibition is all-to-all (*Lin et al., 2014*). The finding of all-to-all inhibition was based on findings that whereas blocking synaptic output from all Kenyon cells vastly increases Kenyon cell odor responses (by

blocking negative feedback via APL), blocking output from one subset of Kenyon cells has no effect on (for α′β′ and γ Kenyon cells), or only weakly increases (for αβ Kenyon cells), odor responses of that subset (*Lin et al., 2014*). This finding is consistent with our findings in *Figure 8*, which predict only preferential, not exclusive, self-inhibition. For example, while blocking output from only γ Kenyon cells would remove γ-to-γ inhibition, γ Kenyon cells (only 1/3 of all Kenyon cells) should still get enough lateral inhibition from the other 2/3 of Kenyon cells to suppress their activity to normal levels.

We based our phenomenological model on the heuristic that depolarization decays exponentially with distance. The model is constrained by experimental data, although these are necessarily imperfect; potential sources of error include automated segmentation and synapse annotation in the connectome, imperfect registration of the hemibrain APL to our standard backbone skeleton, and noisy red dye signals that may not perfectly report ATP localization (see above). We ignored certain complicating factors in our model for simplicity – some due to lack of data (e.g. active conductances; differences in synaptic strength or membrane conductance across APL), others as simply beyond the scope of this study (e.g. dynamic effects of feedback inhibition; effects of neurite branching and differing branch lengths). Some of these neglected factors (e.g. active conducances) could explain why an unusually low ratio of $\frac{R_m}{R_a}$ (0.025 m) is required to fit our experimental measurements of activity spread in APL. Future detailed compartmental modeling will allow further insights into local feedback inhibition in the mushroom body.

Our findings show that localized activity within APL has two broader implications for mushroom body function. First, local activity in APL leads to stronger inhibition of Kenyon cells nearby than far away, to the point that different compartments of the mushroom body lobes defined by dopaminergic and output neuron innervation can receive different inhibition. This local inhibition would allow the single APL neuron to function effectively as multiple inhibitory interneurons, much like mammalian and fly amacrine cells (*Grimes et al., 2010*; *Meier and Borst, 2019*). Each 'sub-neuron' of APL could locally modulate the function of one mushroom body 'compartment', that is, a unit of dopaminergic neurons/Kenyon cells / mushroom body output neurons. Different compartments are innervated by different dopaminergic neurons (e.g. reward vs. punishment) and different mushroom body output neurons (e.g. avoid vs. approach) (*Aso et al., 2014*), and they govern synaptic plasticity and hence learning by different rules (e.g. different speeds of learning/forgetting) (*Aso and Rubin, 2016*). Thus, APL could locally modulate different compartment-specific aspects of olfactory learning, especially given that different regions of APL respond differently to dopamine (*Zhou et al., 2019*) and electric shock punishment (*Figure 1*). If such local inhibition is important for learning, the fact that spatial attenuation of APL's inhibitory effect is gradual and incomplete could explain why mushroom body compartments are arranged in their particular order, with reward and punishment dopaminergic neurons segregated into the horizontal and vertical lobes, respectively. Under this scenario, APL would serve two distinct, spatially segregated functions: enforcing Kenyon cell sparse coding in the calyx, and modulating learning in the compartments of the lobes.

Second, the finding of disproportionate self-inhibition compared to other-inhibition provides a new perspective on APL's function. Inhibition from APL sparsens and decorrelates Kenyon cell odor responses to enhance learned odor discrimination (*Lin et al., 2014*) but in general, decorrelation is better served by all-to-all lateral inhibition than by self-inhibition (see Appendix 1). Self-inhibition does help decorrelate population activity by pushing some neurons' activity below spiking threshold (Appendix 1); this could occur if APL can be activated by subthreshold activity in Kenyon cells, for example, from KC-APL synapses in Kenyon cell dendrites. However, this effect of self-inhibition is better thought of, not as decorrelation per se, but rather as gain control (*Asahina et al., 2009*; *Olsen et al., 2010*; *Root et al., 2008*), effectively equivalent to adjusting the threshold or gain of excitation according to the strength of stimulus. Of course, lateral inhibition is still a strong force in the mushroom body: given that there are ~2000 Kenyon cells, the sum total lateral inhibition that an individual Kenyon cell receives would still be stronger than its own self-inhibition, even with the imbalance predicted by *Figure 8*. Why might the predicted 'bonus' self-inhibition be useful? Beyond its role in sparse coding, APL inhibition is also thought to function as a gating mechanism to suppress olfactory learning (*Liu and Davis, 2009*; *Zhou et al., 2019*); for such a function it would make sense for Kenyon cells to disproportionately inhibit themselves. Future work will address how lateral inhibition interacts with other functions for APL in this local feedback circuit.

# Materials and methods

## Key resources table

| Reagent type (species) or resource | Designation | Source or reference | Identifiers | Additional information |
|---|---|---|---|---|
| Genetic reagent (*D. melanogaster*) | OK107-GAL4 | *Connolly et al., 1996* | BDSC:854 | |
| Genetic reagent (*D. melanogaster*) | MB247-DsRed | *Riemensperger et al., 2005*, | FLYB:FBtp0022384 | |
| Genetic reagent (*D. melanogaster*) | UAS-GCaMP6f (VK00005) | *Chen et al., 2013*, | BDSC:52869 FLYB: FBst0052869 | |
| Genetic reagent (*D. melanogaster*) | UAS-GCaMP6f (attP40) | *Chen et al., 2013*, | BDSC:42747 FLYB: FBst0042747 | |
| Genetic reagent (*D. melanogaster*) | tub-FRT-GAL80-FRT | *Gordon and Scott, 2009*; *Lin et al., 2014* | BDSC:38880 | |
| Genetic reagent (*D. melanogaster*) | MB247-LexA | *Lin et al., 2014*; *Pitman et al., 2011* | FLYB:FBtp0070099 | |
| Genetic reagent (*D. melanogaster*) | LexAop-P2X2 | This study | | See 'Fly strains and husbandry' |
| Genetic reagent (*D. melanogaster*) | LexAop-GCaMP6f | *Barnstedt et al., 2016* | | Gift from S. Waddell |
| Genetic reagent (*D. melanogaster*) | GH146-FLP | *Hong et al., 2009*, | FLYB:FBtp0053491 | |
| Genetic reagent (*D. melanogaster*) | UAS-mCherry-CAAX | *Kakihara et al., 2008*; *Lin et al., 2014* | FLYB:FBtp0041366 | |
| Genetic reagent (*D. melanogaster*) | NP2631-GAL4 | *Tanaka et al., 2008* | DGRC:104266 | |
| Genetic reagent (*D. melanogaster*) | UAS-P2X2 | *Lima and Miesenböck, 2005* | BDSC:76032 FLYB: FBtp0021869 | |
| Genetic reagent (*D. melanogaster*) | UAS-P2X2 (attP40) | *Clowney et al., 2015* | | Gift from V. Ruta |
| Genetic reagent (*D. melanogaster*) | VT43924-GAL4.2 | This study | | See 'Fly strains and husbandry' |
| Genetic reagent (*D. melanogaster*) | UAS-CD8::GFP | *Lee et al., 1999* | BDSC:5130 FLYB: FBst0005130 | |
| Genetic reagent (*D. melanogaster*) | 474-GAL4 | *Silies et al., 2013* | BDSC:63344 FLYB: FBst0063344 | |
| Genetic reagent (*D. melanogaster*) | 853-GAL4 | *Silies et al., 2013* {Gao:2015fk} | InSITE database 0853 | |
| Genetic reagent (*D. melanogaster*) | mb247-GAL4 | *Zars et al., 2000* | | |
| Genetic reagent (*D. melanogaster*) | c739-GAL4 | *Yang et al., 1995*, | BDSC:7362 FLYB: FBst0007362 | |
| Genetic reagent (*D. melanogaster*) | NP3061-GAL4 | *Tanaka et al., 2008*, | DGRC:104360 FLYB:FBti0035130 | |
| Genetic reagent (*D. melanogaster*) | QUAS-GCaMP3 | This study | | See 'Fly strains and husbandry' |
| Genetic reagent (*D. melanogaster*) | GH146-QF | *Potter et al., 2010* | BDSC:30015 FLYB: FBti0129854 | |
| Genetic reagent (*D. melanogaster*) | UAS-Ort | *Liu and Wilson, 2013* | | Gift from Chi-hon Lee |

## Fly strains and husbandry

Flies were kept at 25 °C (experimental) or 18 °C (long-term stock maintenance) on a 12 hr/12 hr light/dark cycle in vials containing 80 g medium cornmeal, 18 g dried yeast, 10 g soya flour, 80 g

malt extract, 40 g molasses, 8 g agar, 25 ml 10% nipagin in ethanol, and 4 ml propionic acid per 1 L water. Details of fly strains are given in the Key Resources Table.

VT43924-Gal4.2 was created via LR Clonase reaction between pENTR-VT43924 and pBPGA-L4.2Uw-2, in which the Gal4 sequence was codon-optimized for *Drosophila* and the *hsp70* terminator was replaced with the SV40 terminator (*Pfeiffer et al., 2010*). The resulting construct was inserted in *attP2* by the University of Cambridge Fly Facility Microinjection Service. lexAop-P2X2 was created by subcloning P2X2 from pUAST-P2X2 (*Lima and Miesenböck, 2005*) into pLOT (*Lai and Lee, 2006*). QUAS-GCaMP3 was created by subcloning GCaMP3 (*Tian et al., 2009*) into pQUAST (*Potter et al., 2010*). pLOT-P2X2 and pQUAS-GCaMP3 were injected by Genetic Services, Inc.

## Odor delivery

Odors at $10^{-2}$ dilution were delivered by switching mass-flow controlled carrier and stimulus streams (Sensirion) via solenoid valves (The Lee Company) controlled by LabVIEW 2015 software (National Instruments). Air flow was 0.5 L/min at the exit of the odor tube, which was positioned ~1 cm from the fly's head. Odor pulses lasted 5 s. At the end of each recording where odor pulses were given, the tubes were cleaned by passing pure air through. A tube connected to a vacuum pump was positioned behind the fly to deplete the air of odors remaining after the odor pulse.

## Drug delivery

ATP or GABA at the concentrations indicated in the figure legends, together with a red dye to visualize fluid ejection (SeTau-647, SETA BioMedicals K9-4149, [*Podgorski et al., 2012*]), were applied locally by pressure ejection from patch pipettes (resistance ~10 MΩ; capillary inner diameter 0.86 mm, outer diameter 1.5 mm - Harvard Apparatus 30–0057) coupled to a Picospritzer III (Parker) (puff duration 10 ms, pressure 12.5 psi unless otherwise indicated), which was triggered by the same software as odor delivery. The patch pipette was mounted on a micromanipulator (Patchstar, Scientifica) and positioned at the tip of the vertical lobe, close to the junction point in the horizontal lobe, or in the central part of the calyx. Before initiating recording, fluid was manually ejected to ensure that the pipette was not blocked. For experiments where ATP or GABA were applied during odor stimulus, the fluid ejection was triggered at the same time as the odor pulse, although the odor arrived slightly later due to the travel time of the air stream. Histamine (2 mM, Sigma H7250) was added to the bath 5 min before imaging in APL>Ort experiments.

## Electric shock

A rectangle of stacked copper plates was brought into contact with the fly's abdomen and legs using a manually movable stage (DT12XYZ/M, Thorlabs). The design of the copper plates was based on *Felsenberg et al., 2018*. Shocks were provided by a DS3 Constant Current Isolated Stimulator (1.2 s, 32 mA, Digitimer; maximum voltage 90 V) and triggered by the same software as the odor delivery. Successful shock delivery was confirmed by observing the fly's physical reaction with a Genie Nano-M1280 camera (Teledyne Dalsa) coupled to a 1x lens with working distance 67 mm (SE-16SM1, CCS).

## Functional imaging

Flies were cold-anesthetized and mounted using wax and dental floss in a hole in a piece of aluminum foil fixed to a perfusion chamber, such that the fly's dorsal and ventral sides were kept on opposite sides of the foil. The dorsal part was immersed in carbogenated (95% $O_2$, 5% $CO_2$) external solution (103 mM NaCl, 3 mM KCl, 5 mM trehalose, 10 mM glucose, 26 mM NaHCO$_3$, 1 mM NaH$_2$PO$_4$, 3 mM CaCl$_2$, 4 mM MgCl2, 5 mM TES, pH 7.3). The cuticle overlying the brain was carefully removed using forceps, followed by removal of fat tissue and trachea. During experiments, the brain was continuously perfused with carbogenated external solution (1.96 ml/min) using a Watson-Marlow pump (120S DM2).

Brains were initially inspected using widefield microscopy (Moveable Objective Microscope, Sutter) and a xenon-arc lamp (LAMBDA LS, Sutter). Functional imaging was carried out using two-photon laser-scanning microscopy (*Ng et al., 2002*; *Wang et al., 2003*). Fluorescence was excited by a Ti-Sapphire laser (Mai Tai eHP DS, Spectra-Physics; 80 MHz, 75–80 fs pulses) set to 910 nm, which was attenuated by a Pockels cell (350-80LA, Conoptics) and steered by a galvo-resonant scanner

(RESSCAN-MOM, Sutter). Excitation was focussed using a 1.0 NA 20X objective (XLUMPLFLN20XW, Olympus). Emitted light passed through a 750 nm short pass filter (to exclude excitation light) and bandpass filters (green: 525/50; red: 605/70), and was detected by GaAsP photomultiplier tubes (H10770PA-40SEL, Hamamatsu Photonics) whose currents were amplified (TIA-60, Thorlabs) and transferred to the imaging computer, which ran ScanImage 5 (Vidrio). Volume imaging was carried out using a piezo objective stage (nPFocus400, nPoint).

Results in *Figure 3* were generated similarly but on different equipment as described in *Lin et al., 2014*: Movable Objective Microscope with galvo-galvo scanners, Chameleon Ultra II laser (Coherent), 20x, 1.0 NA W-Plan-Apochromat objective (Zeiss), HCA-4M-500K-C current amplifier (Laser Components), MPScope 2.0 software (Sutter). Flies were heated with a TC-10 temperature controller (NPI) and HPT-2 inline perfusion heater (ALA). The temperature at the fly was measured with a TS-200 temperature sensor (NPI) and a USB-1208FS DAQ device (Measurement Computing) at 30 Hz; temperature traces were smoothed over 20 frames by a moving-average filter to remove digitization artifacts.

## Structural imaging

Dissected VT43924-GAL4.2>GFP brains were fixed in 4% (wt/vol) paraformaldehyde in PBT (100 mM $Na_2PO_4$, 0.3% Triton-X-100, pH 7.2), washed in PBT (two quick washes, then three 20 min washes) and mounted in Vectashield. Confocal stacks were taken on a Nikon A1 confocal microscope in the Wolfson Light Microscopy Facility. Unfixed brains were mounted in PBS using two stacked reinforcement rings (Avery 5722) as spacers and imaged on the two-photon.

## Image analysis

Motion correction was carried out as in *Bielopolski et al., 2019*; *Lin et al., 2014*; flies with excessive uncorrectable motion were discarded. For experiments without 3D-skeletonization (see below), $\Delta F/F$ was calculated in ImageJ for manually drawn regions of interest (ROIs). Background fluorescence was taken from an empty region in the deepest z-slice, and subtracted from the ROI fluorescence. $F_0$ was calculated as the average fluorescence during the pre-stimulus period and $\Delta F/F$ was calculated as $(F-F_0)/F_0$. $\Delta F/F$ data were smoothed by a boxcar filter of five frames (*Figure 3*) or 1 s (*Figures 1*, *4*, *5C–E*, and *6*) for displaying time course traces or calculating the peak response, but not for calculating the average response (*Figures 5F–H* and *7*).

## 3D-skeletonization

Where the same neuron was recorded in multiple conditions, movies were aligned by maximizing cross-correlation between the time-averaged mb247-dsRed or mb247-LexA > GCaMP6f signals of each recording. This aligned image was used to define the 3D volume of the mushroom body neuropil by manually outlining it in each z-slice (excluding Kenyon cell somata). This volume mask was converted to a 'backbone skeleton' passing through the center of the major branches of the mushroom body (vertical lobe, horizontal lobe, peduncle/calyx), by manually defining key points (e.g. tip of the vertical/horizontal lobes, junction, places where the structure curves) and connecting them into an undirected graph.

To compare backbones across different flies, a 'standard' APL backbone was generated by taking the average length of each branch across all flies (*Figure 5*). Each individual backbone was normalized to this standard backbone by placing nodes at spacing x*20 μm, where x is the ratio of the branch length of the individual backbone over the branch length of the standard backbone. The 3D volume of each APL neuron was partitioned into Voronoi cells, each with one of the evenly spaced backbone nodes as the centroid. For each node, fluorescence signal at each time point was averaged over all voxels in the corresponding Voronoi cell. The background was subtracted and baseline fluorescence calculated as the average fluorescence during the pre-stimulus period. $\Delta F/F$ was calculated for each node and time point as the difference between the fluorescence at that time point and the baseline fluorescence, divided by the baseline fluorescence.

$\Delta F/F$ at each node was calculated as both the time-average over the stimulus period (for individual flies), and as the average across flies within a group, to show the time course. In the latter case, time courses were linearly interpolated to a frame time of 0.2 s. $\Delta F/F$ vs. distance was plotted according to the normalized distance of each node from the dorsal calyx on the 'standard'

backbone. Individual nodes were excluded if the standard deviation of ΔF/F during the pre-stimulus period was greater than 1.0 (green channel), or 0.7 (red channel), as such high noise indicated poor signal in that segment. Normalized inhibitory effect (of ATP or GABA) was calculated as (ΔF/F odor with ATP/GABA - ΔF/F odor alone) divided by the peak ΔF/F for odor alone. We divided by peak rather than average odor response because this was more robust for recordings where the odor response was low, where normalizing to the average caused extreme magnification of small changes.

## Connectome analysis

We reasoned that the strength of lateral inhibition of one Kenyon cell (KC1) onto another (KC2) could be modeled as

$$s(k_1, k_2) = \sum_{i=1}^{m} \sum_{j=1}^{n} w\left(x_{i,k_1}, y_{j,k_2}\right) \tag{1}$$

where $x_{i,k_1}$ is the $i$th synapse from KC1 onto APL, and $y_{j,k_2}$ is the $j$th synapse from APL onto KC2, and $w$ is given by

$$w\left(x_{i,k_1}, y_{j,k_2}\right) = e^{-d\left(x_{i,k_1}, y_{j,k_2}\right)/\lambda} \tag{2}$$

where $d$ is the distance between the two synapses along the APL neurite skeleton and $\lambda$ is the space constant. We used *Equation 2* when treating all neurites as having the same radius. Because APL neurites have varying widths along their length, and the space constant is proportional to the square root of the neurite radius, every neurite segment potentially has a different space constant. If the neurite between $x_{i,k_1}$ and $y_{j,k_2}$ is composed of $n$ segments, where the $i$th segment is a cylinder of length $d_i$ and radius $r_i$, with a space constant $\lambda_i$ proportional to the square root of $r_i$, then *Equation 2* becomes $w\left(x_{i,k_1}, y_{j,k_2}\right) = \prod_{i=1}^{n} e^{-\frac{d_i}{\lambda_i}} = e^{-\sum_{i=1}^{n} \frac{d_i}{\lambda_i}} = e^{-\sum_{i=1}^{n} \frac{d_i}{k\sqrt{r_i}}} = e^{-\frac{1}{k}\sum_{i=1}^{n} \frac{d_i}{\sqrt{r_i}}}$, where $\frac{d_i}{\sqrt{r_i}}$ can be considered the 'electrotonic' length of each segment, that is, the length normalized by the square root of the radius (wider segments are 'shorter' in terms of signal decay than thinner segments), and $k$ is the space constant in 'electrotonic' space ($k$ depends on the specific axial and membrane resistance of the APL neuron; see below). However, in the hemibrain connectome, radius is a property of nodes, not a property of segments; thus, for the ~50% of segments in APL where the two nodes at each end have different radii, the segment is a truncated cone, not a cylinder (*Scheffer et al., 2020*). For a truncated cone where the radii at the two ends are $r_{i,1}$ and $r_{i,2}$, the 'electrotonic' length is not $\frac{d_i}{\sqrt{r_i}}$ but rather $\int_0^{d_i} \frac{dx}{\sqrt{\frac{r_{i,2}-r_{i,1}}{d_i}x + r_{i,1}}}$, or $2d_i \frac{\sqrt{r_{i,2}} - \sqrt{r_{i,1}}}{r_{i,2} - r_{i,1}}$ (note that when $r_{i,2} = r_{i,1}$, this reduces to $\frac{d_i}{\sqrt{r_i}}$ by L'Hôpital's rule). For synapses falling on the same truncated cone segment, at locations $x_1$ and $x_2$ as measured by distance from the node with radius $r_{i,1}$, the electrotonic distance is $\int_{x_1}^{x_2} \frac{dx}{\sqrt{\frac{r_{i,2}-r_{i,1}}{d_i}x + r_{i,1}}}$ or $\frac{2d_i \left(\sqrt{\frac{r_{i,2}-r_{i,1}}{d_i}x_2 + r_{i,1}} - \sqrt{\frac{r_{i,2}-r_{i,1}}{d_i}x_1 + r_{i,1}}\right)}{r_{i,2} - r_{i,1}}$ (when $r_{i,2} = r_{i,1}$, this reduces to $\frac{x_2 - x_1}{\sqrt{r_i}}$; when $x_1 = 0$ and $x_2 = d_i$, this reduces to $2d_i \frac{\sqrt{r_{i,2}} - \sqrt{r_{i,1}}}{r_{i,2} - r_{i,1}}$). Thus, *Equation 2* becomes

$$w\left(x_{i,k_1}, y_{j,k_2}\right) = e^{-\frac{1}{k}\sum_{i=1}^{n} \frac{2d_i\left(\sqrt{\frac{r_{i,2}-r_{i,1}}{d_i}x_{i,2} + r_{i,1}} - \sqrt{\frac{r_{i,2}-r_{i,1}}{d_i}x_{i,1} + r_{i,1}}\right)}{r_{i,2} - r_{i,1}}} \tag{3}$$

To relate $k$ to the average space constant in 'real' space (i.e. $\lambda$), consider that the decay over the whole distance between $x_{i,k_1}$ and $y_{j,k_2}$ is $w\left(x_{i,k_1}, y_{j,k_2}\right) = e^{-\frac{\sum_{i=1}^{n} d_i}{\lambda}}$. Therefore, $\frac{1}{k}\sum_{i=1}^{n} 2d_i \frac{\sqrt{r_{i,2}} - \sqrt{r_{i,1}}}{r_{i,2} - r_{i,1}} = \frac{1}{\lambda}\sum_{i=1}^{n} d_i$, so

$\lambda = k \frac{\sum_{i=1}^{n} d_i}{\sum_{i=1}^{n} 2d_i \frac{\sqrt{r_{i,2}} - \sqrt{r_{i,1}}}{r_{i,2} - r_{i,1}}}$, or $k$ multiplied by the real length divided by the 'electrotonic' length. Over the entire APL neuron, the total length of neurites ($\sum_{i=1}^{n} d_i$) is 0.08018 m (1.0022 x $10^7$ pixels; 1 pixel = 8

nm) while the total 'electrotonic' length ($\sum\limits_{i=1}^{n} 2d_i \frac{\sqrt{r_{i,2}} - \sqrt{r_{i,1}}}{r_{i,2} - r_{i,1}}$) is 179 m$^{1/2}$ (2.0019 x 10$^6$ pixels$^{1/2}$), so $\frac{\lambda}{k}$ = 4.48 x 10$^{-4}$ m$^{1/2}$ (5.006 pixels$^{1/2}$). Note that this value is close, but not equal, to the square root of the average radius (0.247 μm, or 30.96 pixels). Thus, for example, when we report that we tested $\lambda$ = 50 μm (6250 pixels), this corresponds to $k$ = 0.1117 m$^{1/2}$ (1249 pixels$^{1/2}$).

Given that we defined $\lambda = k\sqrt{r}$, $k$ can be calculated from the equation $\lambda = \sqrt{\frac{rR_m}{2R_i}}$ where $R_m$ is the specific membrane resistance and $R_a$ is the specific axial (internal) resistance. Reported ratios of $\frac{R_m}{R_a}$ in *Drosophila* neurons range from 0.0907 m ($R_m$ = 0.8166 Ωm$^2$, $R_a$ = 9 Ωm in HS cells [*Cuntz et al., 2013*], giving $k$ = 0.21 m$^{1/2}$ and $\lambda$ = 95.4 μm) to 2.0 m ($R_m$ = 2.04 Ωm$^2$, $R_a$ = 1.02 Ωm in one projection neuron [*Gouwens and Wilson, 2009*], giving $k$ = 1 m$^{1/2}$ and $\lambda$ = 447 μm). To simulate these and other space constants, we placed ~10,000 points at random on APL's neurite skeleton. We divided the backbone skeleton (*Figure 5*) into 10 μm segments. We simulated the activity in segment $s_i$ ($i$ = 1...35) as the count of every pair between points in $s_i$ and points in the whole APL, weighted by the distance separating the pair:

$$activity(s_i) = \frac{1}{N_{s_i}} \sum_{j \in s_i}^{N_{s_i}} \sum_{k=1}^{n} w(p_j, p_k) \tag{4}$$

where $w(p_j, p_k)$ depends on the distance between points $p_j$ and $p_k$ (as in *Equation 3*, or *Equation 2* when ignoring neurite radius), $N_{s_i}$ is the number of points ($j \in s_i$) lying within segment $s_i$, and $n$ = the number of random points (~10,000). We excluded points outside the mushroom body (e.g. the neurite leading to the APL soma).

To calculate the distance from every KC-APL synapse to every APL-KC synapse (setting no lower threshold on confidence of synapse annotation), we calculated the distance from each APL-KC synapse to its nearest neighbor APL-KC synapses, as measured along the skeleton of APL neurites. We used Dijkstra's algorithm on the resulting undirected graph to calculate the distance from each APL-KC synapse to every other APL-KC synapse. We then calculated the distance from each KC-APL synapse to its nearest neighbor APL-KC synapses, and combined this with the distances between APL-KC synapses to find the distance from every KC-APL synapse to every APL-KC synapse. To limit analysis to APL-KC synapses in the calyx, we used APL-KC synapses whose pixel location had a y-value <20000 (160 μm) (i.e. posterior to the peduncle) and that were not annotated as being in the PED(R) ROI (*Scheffer et al., 2020*), because some APL-KC synapses, especially those in the accessory calyces, are annotated as being outside the CA(R) ROI.

## Analysis of RNA-seq data

Transcripts per million (TPM) values (*Aso et al., 2019*) were summed arithmetically across splice variants for each gene. The log$_{10}$ of each resulting value was taken. To prevent negative infinities, we set the log$_{10}$ of zero values to be −2 (i.e. TPM = 0.01), because the minimum TPM value in the dataset was 0.1. The arithmetic mean of log$_{10}$(TPM) was taken across biological replicates for each cell type (i.e. geometric mean of TPM). Gene Ontology (*Ashburner et al., 2000*; *The Gene Ontology Consortium, 2019*) was used to categorize by biological process the 1529 genes in which APL's TPM counts were higher than all 20 other cell types in the dataset. The list of ion channels was curated from The Interactive Fly, FlyBase and (*Groschner et al., 2018*; *Walcott et al., 2018*); genes whose mean TPM across all non-APL MB neurons was less than one were excluded.

## Statistics and software

Statistical analysis and curve fitting were carried out in Prism 8 (GraphPad). Parametric (t-test, ANOVA) or non-parametric tests (Wilcoxon, Mann-Whitney, Friedman, Kruskal-Wallis) were used depending on whether data passed the D'Agostino-Pearson (or Shapiro-Wilk, for small n) normality test. Traces of manual ROIs were analysed in Igor Pro 7 (WaveMetrics). 3D-skeletonization and analysis of connectome and RNA-seq data was carried out with custom software written in Matlab (MathWorks), which is available at http://github.com/aclinlab. Hemibrain connectome data (v1.1) was downloaded using neuprint-python (https://github.com/connectome-neuprint/neuprint-python). No explicit power analysis was used to pre-determine sample sizes; we used sample sizes comparable

to those used in similar studies (e.g. *Zhou et al., 2019*). The experimenter was not blind to experimental conditions or genotypes.

## Acknowledgements

We thank Moshe Parnas, Przemyslaw Stempor, Natalia Bulgakova, and members of the Lin and Juusola labs for discussions, Yoshinori Aso for sharing the RNA-seq data before publication, Saket Navlakha for help with accessing the hemibrain data, Mark Kelly for preliminary analyses of data from *Takemura et al., 2017*, Emmanuel Perisse for the design of the electric shocker, Scott Waddell, Chihon Lee, Vanessa Ruta, the Bloomington Stock Center, and the Vienna *Drosophila* Resource Center for fly stocks, and Lily Bolsover, Chloe Donahue, Kath Whitley, Josh Marston, Rachael Thomas, and Rachid Achour for technical assistance. This work was supported by the European Research Council (639489) and the Biotechnology and Biological Sciences Research Council (BB/S016031/1).

## Additional information

### Funding

| Funder | Grant reference number | Author |
|---|---|---|
| H2020 European Research Council | 639489 | Andrew C Lin |
| Biotechnology and Biological Sciences Research Council | BB/S016031/1 | Andrew C Lin |

The funders had no role in study design, data collection and interpretation, or the decision to submit the work for publication.

### Author contributions

Hoger Amin, Conceptualization, Software, Formal analysis, Investigation, Visualization, Methodology, Writing - original draft, Writing - review and editing; Anthi A Apostolopoulou, Raquel Suárez-Grimalt, Investigation, Writing - review and editing; Eleftheria Vrontou, Resources, Writing - review and editing; Andrew C Lin, Conceptualization, Software, Formal analysis, Supervision, Funding acquisition, Investigation, Visualization, Methodology, Writing - original draft, Writing - review and editing

### Author ORCIDs

Hoger Amin  https://orcid.org/0000-0002-7884-4815
Anthi A Apostolopoulou  http://orcid.org/0000-0002-8174-4372
Raquel Suárez-Grimalt  https://orcid.org/0000-0002-5374-7963
Andrew C Lin  https://orcid.org/0000-0001-6310-9765

### Decision letter and Author response

Decision letter https://doi.org/10.7554/eLife.56954.sa1
Author response https://doi.org/10.7554/eLife.56954.sa2

## Additional files

### Supplementary files

- Supplementary file 1. List of full genotypes.
- Supplementary file 2. Details of statistical analysis.
- Transparent reporting form

### Data availability

All data generated or analyzed during this study are included in the manuscript and supporting files. Source data files have been provided for Figures 1, 2, 2-supplement 1, 3, 4, 7 (Note Fig. 7 includes

data from Fig. 5), 7-supplement 1, 8, 8-supplements 2&3. Custom software is available at https://github.com/aclinlab/amin-et-al-2020 and https://github.com/aclinlab/calcium-imaging.

The following previously published dataset was used:

| Author(s) | Year | Dataset title | Dataset URL | Database and Identifier |
|---|---|---|---|---|
| Aso Y, Ray RP, Long X, Bushey D, Cichewicz K, Ngo TT, Sharp B, Christoforou C, Hu A, Lemire AL, Tillberg P, Hirsh J, Litwin-Kumar A, Rubin GM | 2019 | Bulk RNA-seq data from the dopaminergic neurons, MB output neurons, Kenyon cells, APL and DPM neurons in adult *Drosophila* | https://www.ncbi.nlm.nih.gov/geo/query/acc.cgi?acc=GSE139889 | NCBI Gene Expression Omnibus, GSE139889 |

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

## Appendix 1

### Effect of self-inhibition vs. all-to-all inhibition on decorrelation of population activity

Here we develop and formalize our intuition for why all-to-all lateral inhibition is better than self-inhibition for decorrelating population activity, that is, for enhancing contrast between stimuli.

Consider two linear rate-coding neurons such that the population response of the two neurons can be described as a vector $\mathbf{y} = (y_1, y_2)$, e.g., if neuron 1's activity is 6 and neuron 2's is 7, then population activity $\mathbf{y} = (6, 7)$. Take a simple case where $\mathbf{x} = (x_1, x_2)$ is the excitation and $\theta$ represents the spiking threshold:

$$\mathbf{y} = relu(\mathbf{x} - \theta) \tag{1}$$

where relu is a rectified linear unit:

$$relu(x) = \begin{cases} x & \text{if } x \geq 0 \\ 0 & \text{if } x < 0 \end{cases} \tag{2}$$

Suppose the two neurons respond to two similar stimuli, such that $\mathbf{x}^{(1)} = (9, 10)$ and $\mathbf{x}^{(2)} = (10, 9)$. If $\theta = 3$, then $\mathbf{y}^{(1)} = (6, 7), \mathbf{y}^{(2)} = (7, 6)$ (plotted on *Appendix 1—figure 1A*).

Consider a form of inhibition in which the principal neurons excite the inhibitory neuron from their dendrites, not necessarily needing to depolarize above the spiking threshold. This is similar to the mushroom body, where Kenyon cells release acetylcholine from their dendrites and locally activate the inhibitory APL. If each principal neuron inhibits only itself by local inhibition - which we call self-inhibition - then

$$\mathbf{y} = relu(\mathbf{x} - \alpha \mathbf{x} - \theta) = relu((1 - \alpha)\mathbf{x} - \theta) \tag{3}$$

where $\alpha$ is the gain on the inhibitory neuron ($0 < \alpha < 1$). Here, the self-inhibition is equivalent to decreasing the gain on the excitation. If we set $\alpha = 0.5$ and $\theta = 3$, then $\mathbf{y}^{(1)} = (1.5, 2)$ and $\mathbf{y}^{(2)} = (2, 1.5)$ (*Appendix 1—figure 1A*). In contrast, if the inhibitory neuron adds up the activity of all principal neurons equally - which we call all-to-all inhibition - then

$$\mathbf{y} = relu(\mathbf{x} - \alpha \langle \mathbf{x} \rangle - \theta) \tag{4}$$

where $\langle \mathbf{x} \rangle$ is the average excitation across the population. Using the same parameters as for self-inhibition above, $\mathbf{y}^{(1)} = (1.25, 2.25)$ and $\mathbf{y}^{(2)} = (2.25, 1.25)$ (*Appendix 1—figure 1A*).

We quantify the separation in the population response to these two stimuli using the cosine distance, that is, $1 - \cos(\phi)$, where $\phi$ is the angle between the two vectors $\mathbf{y}^{(1)}$ and $\mathbf{y}^{(2)}$. It can be seen in *Appendix 1—figure 1A* that self-inhibition increases the angle between the two vectors, but all-to-all inhibition increases it even more.

We generalize this example to a wider range of pairs of stimuli:

$$\mathbf{x}^{(1)} = (x_0, ax_0), \mathbf{x}^{(2)} = (ax_0, x_0) \tag{5}$$

where $x_0$ ranges from 1 to 20, $a$ from 0.1 to 0.9. This generates pairs of stimuli that are very different when $a$ is small, e.g., (1.5, 15) and (15, 1.5), or that are very similar when $a$ is large, e.g. (13.5, 15) and (15, 13.5). We set $\theta = 2$.

In the no-inhibition case, cosine distances are higher at lower $x_0$ because this brings activity closer to the threshold (*Appendix 1—figure 1B*). That is, (3, 4) and (4, 3) become more different from each other once a threshold of 2 is applied, turning the vectors to (1, 2) and (2, 1), but the effect is not so strong for turning (13, 14) and (14, 13) into (11, 12) and (12, 11).

Compared to the no-inhibition case, self-inhibition increases cosine distances for a given $x_0$ and $a$. However, this is functionally equivalent to simply raising $\theta$ dynamically for higher $x_0$. Visually, this is seen on *Appendix 1—figure 1B* by the way the self-inhibition panel is simply the no-inhibition panel stretched vertically. That is, self-inhibition decorrelates mainly by using gain control to bring down the activity closer to threshold.

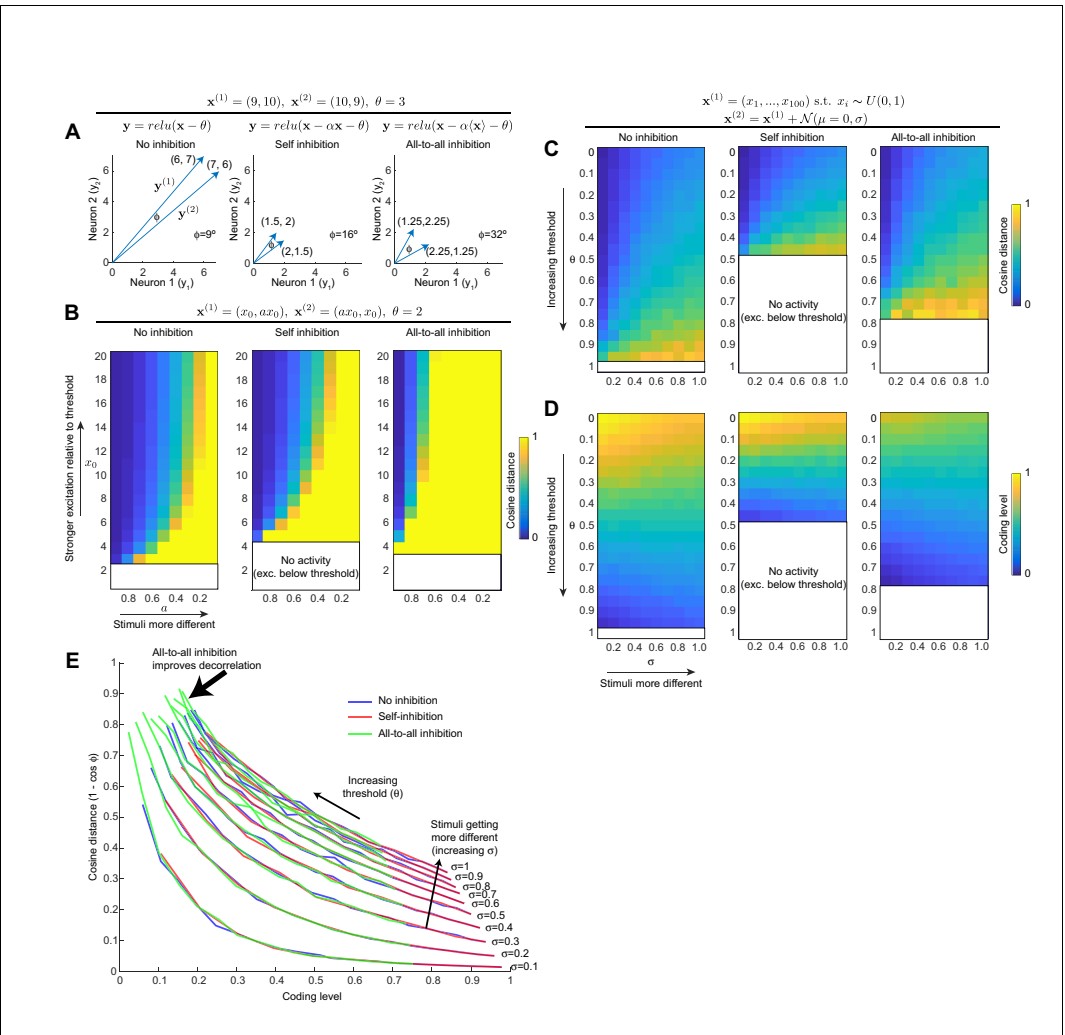

**Appendix 1—figure 1.** All-to-all inhibition decorrelates population activity better than self-inhibition. (**A**) For a toy example of two neurons and two stimuli, self-inhibition improves the stimulus separation (i.e, increases angle $\phi$ between the two stimulus vectors) by bringing activity down closer to spiking threshold, but all-to-all inhibition improves the separation more. Note that to simplify the model, we model the inhibitory interneuron as being activated by the principal neurons' dendrites, as occurs with Kenyon cells and APL. (**B**) Generalization of result in (**A**) to wider activity range in the two neurons. Heat maps show cosine distance for different values of $x_0$ and $a$ (increasing $x_0$ means more excitation; decreasing $a$ means the stimuli are more different). The self-inhibition panel is the same as the no-inhibition panel, just stretched vertically (i.e., self-inhibition acts as gain control), whereas the all-to-all inhibition panel expands the range of high cosine distances toward more similar stimulus pairs. Blank areas mean the activity is zero because excitation is below threshold, so cosine distance is undefined. (**C**) Generalization of (**A,B**) to population of 100 neurons. Stimulus 1 is sampled from a uniform distribution over (0, 1). Stimulus 2 is Stimulus 1 plus Gaussian noise (magnitude of noise governed by $\sigma$; higher $\sigma$ means the two stimuli are more different). Heat maps show cosine distance for different thresholds ($\theta$) and $\sigma$ averaged over 100 trials. (**D**) Coding level (fraction of active neurons) for different thresholds and $\sigma$ as in (**C**); note how decreasing coding level (increased sparseness) correlates with higher cosine distance. (**E**) Each curve is one column (one value of $\sigma$) from the heat maps in (**C**) and (**D**), showing how cosine distance improves with lower coding levels. Adding inhibition does not change this curve; it merely shifts the model to different points along the curve. All-to-all inhibition achieves higher cosine distance (more decorrelated activity) than self-inhibition or no inhibition.

In contrast, all-to-all inhibition improves decorrelation for higher $a$ (i.e., more similar stimuli). Visually, this is seen on *Appendix 1—figure 1B* as the zone of large distances expanding to the left. That is, all-to-all inhibition can make similar stimuli appear more distinct.

Next we generalize these intuitions to larger populations of neurons. Consider a population of 100 neurons that receive excitation x:

$$\mathbf{x} = (x_1, ..., x_{100}) \tag{6}$$

Stimulus 1 consists of random numbers between 0 and 1 (uniform distribution):

$$\mathbf{x}^{(1)} = (x_1, ..., x_{100}) \text{ s.t. } x_i \sim U(0, 1) \tag{7}$$

Stimulus 2 is similar to Stimulus 1, which we simulate by making it equal to Stimulus 1 plus Gaussian noise with mean 0 and standard deviation σ:

$$\mathbf{x}^{(2)} = \mathbf{x}^{(1)} + \mathcal{N}(\mu = 0, \sigma) \tag{8}$$

We apply these excitations to the no-inhibition, self-inhibition, or all-to-all inhibition conditions, with θ ranging from 0 to 0.1 and σ ranging from 0.1 to 1. For all three conditions, cosine distances increase for higher σ because the two stimuli are more different (*Appendix 1—figure 1C*). They also increase for higher θ because more neurons are silenced (i.e., the coding level, or fraction of responsive neurons, is reduced) (*Appendix 1—figure 1D*).

Compared to the no-inhibition condition, self-inhibition reaches similar cosine distances but at lower θ, consistent with our earlier conclusion that self-inhibition mainly functions equivalently to dynamically adjusting the threshold. In other words, self-inhibition helps decorrelate by pushing neurons' activity down closer to, or indeed below, the threshold. In contrast, all-to-all inhibition (1) reaches higher cosine distances than the other two conditions and (2) allows higher cosine distances at lower thresholds.

This can be more clearly understood by plotting cosine distance against coding level for different values of σ. In *Appendix 1—figure 1E*, each line is one column (one value of σ) in the heat maps in *Appendix 1—figure 1C,D*. Within this range of parameters, the fundamental relation between coding level and cosine distance is fixed for a given σ. That is, for a given σ, all three conditions follow the same curve on the cosine distance vs. coding level graph. What differs is *where* on the curve they fall. The decorrelating benefit created by all-to-all inhibition arises in part from pushing the population activity up and to the left along the curve. A second benefit arises at the upper left end of the curve: this is where population activity falls silent because θ is so high (corresponding to the blank rows in the heat maps). All-to-all inhibition allows non-zero population activity to persist a bit further on this curve, where the other two conditions have already fallen silent. In contrast, self-inhibition merely moves population activity further along the curve at lower values of θ without moving beyond the range allowed without inhibition.

