## [Decision Letter]

**Acceptance summary:**

The present study addresses for the first time the presence of spatially segregated GABAergic inhibition in the mushroom body, the main learning and memory center in the fly brain. Based on observations of calcium fluxes restricted to distinct subdomains of a widely arborizing, non-spiking, GABAergic neuron (APL), the authors propose that localised/compartmentalized inhibition can arise through local excitation and passive electrical responses in neuronal processes. Although the contribution of this localised inhibition to behaviourally relevant computations or hypothesised functions of the APL in learning, memory or odor coding, remain to be addressed, the observations made here are not only relevant for understanding sensory representation and learning in *Drosophila*, but also point to the potential importance of segregated inhibition by interneurons in neuronal networks in general.

**Decision letter after peer review:**

Thank you for submitting your article "Localized inhibition in the *Drosophila* mushroom body" for consideration by *eLife*. Your article has been reviewed by three peer reviewers, including Mani Ramaswami as the Reviewing Editor and Reviewer #1, and the evaluation has been overseen by Ronald Calabrese as the Senior Editor The following individuals involved in review of your submission have agreed to reveal their identity: Tamara Boto (Reviewer #2).

The reviewers have discussed the reviews with one another and the Reviewing Editor has drafted this decision to help you prepare a revised submission.

Summary:

In their manuscript "Localized inhibition in the *Drosophila* mushroom body", Lin and colleagues seek to make a systematic description of how widely neuronal activity propagates in APL, a giant, non-spiking inhibitory neuron in the fly olfactory circuit. In order to address this question the authors employ single cell mRNA sequencing as well as a battery of tests ranging from sensory stimulation via olfactory and electric shock pathways, genetically driven stimulation via TRP and PTX channels, and modeling work to demonstrate that neural activity is greatly reduced during propagation throughout the neuron, likely because of a deficit in ion channels required for active propagation of electrical potentials. The authors demonstrate how this decrement in signal amplitude and the reciprocal connectivity with the Kenyon cell population of the mushroom body results in the modulatory effects of the APL neuron on KCs becoming substantially localized. Modeling results suggest that, through the GABA-mediated feedback circuit this leads to KC activity from single neurons preferentially inhibiting similar KCs to a greater extent than dissimilar KCs, and to KCs preferentially inhibiting themselves more than any other KCs. This prediction has implications for mechanisms of sensory coding and learning in *Drosophila*, as well as for general appreciation of inhibitory mechanisms used by interneurons in neuronal networks. The manuscript is well-written and planned, makes appropriate use of the tools available in the model organism, and presents an interesting, novel result. However, the authors should address several concerns before the manuscript is ready for publication.

Essential revisions:

1) The most significant issue that needs to be addressed is the extent to which the APL neuron's response is localized and pertain to data in Figure 6 and Figure 7 and their interpretation. Based on Figure 6 and Figure 7, the authors argue that locally activated APL preferentially inhibits local KCs. Consistent with this, the inhibition effect appeared more pronounced in local KCs than distal KCs when APL was locally activated in the horizontal lobe and vertical lobe. However:

(1a) In Figure 6, local activation of APL neuron leads to hyperpolarisation of KCs in other regions as well, in addition to the respective region where APL neuron was activated. This should be explained and discussed.

(1b) In Figure 6C (and Figure 7C), local activation of APL by ATP injection into the calyx leads to depolarisation, rather than inhibition of proximal KCs in calyx region, which could potentially be due to leaky expression of P2X2 channel in KCs (subsection “APL inhibits Kenyon cells mostly locally”). However, this seems unlikely: considering that ATP-evoked activity in KCs is comparable throughout the mushroom body in the negative control flies (Figure 6D), the leaky expression cannot explain the greater inhibition in distal KCs than proximal KCs. Assuming that the excitatory responses of KCs we see from the ATP injection on the calyx (APL) is due to leaky expression of P2X2, it still does not explain the inhibition at the vertical and horizontal lobes, which are away further away from the APL activation site. The inhibition at the vertical and horizontal lobes contradicts panels A and B. This discrepancy appears to contradict one of their main findings that spatially restricted APL activity mostly inhibits KCs locally and needs to be addressed.

(1c) This apparent discrepancy is considered in the Discussion section, speculating that it could be caused by GABA receptors. Again, the observed discrepancy contradicts their main finding that differential spatial inhibition in KCs arises from local activity in APL and the spatial arrangement of KC-APL and APL-KC synapses. The authors need to revisit their explanation for why APL, when locally activated in the calyx, preferentially inhibits distal KCs rather than proximal ones.

(1d) Subsection “Local GABA application mimics the inhibitory effect of local APL stimulation”: "strongest at the site of stimulation but still affecting Kenyon cells far from the site of stimulation, to a lesser extent." This statement appears to contradict the assertion that APL provides mainly local inhibition to KCs. The authors should moderate their claims throughout the text to make clear that APL provides both local and more global inhibition.

(1e) If additional experiments are needed to better establish the author's conclusions, then these should be described in the discussion (e.g. an analysis of the odor responses in MBON of neighboring/distal compartments could be important to clarify some conclusions and predictions of the local activation model).

(2) The authors perform a thorough computational analysis of the connectomics data and suggest that each Kenyon cell inhibits itself more than others. This model would support the idea of a spatially restricted APL-mediated inhibition of Kenyon cells, and the authors concluded that "…our physiological measurements of localized activity of APL, combined with the spatial arrangements of KC-APL and APL-KC synapses, predict that Kenyon cells disproportionately inhibit themselves compared to other individual Kenyon cells" (Discussion section). This conclusion contradicts previous results by the same authors. Lin et al., (2014) published not only that the feedback inhibition is from all Kenyon cells to all Kenyon cells, but taken the specific case of the γ lobes of the mushroom bodies, Lin et al. showed the opposite result than the one expected in the present study: blocking synaptic release in γ lobes results in a decrease of the calcium signal of those same γ Kenyon cells, which contradicts the idea of γ KC strongly inhibiting γ Kenyon cells through APL. The manuscript should be revised to include a more thorough discussion regarding these contradictory results. The APL inhibition might vary depending on specific situations; all-to-all feedback might be essential for odor discrimination, whereas a more compartmentalized inhibition might be crucial during reinforcement: APL receives direct spatially-segregated input from dopaminergic neurons (Zhou et al., 2019; and Figure 1 of the present article) and these connections are not included in the connectomic analysis.

(3) Further, regarding Figure 8F-L and connectome analysis in methods, the authors appear to make an error in their choice of length constant. The authors chose length constant = square_root(radius) (subsection “Connectome analysis”), leading them to a conclusion consistent with their experimental observation. However, published physiological data suggest that membrane resistivity should really be about two to three orders of magnitude higher than cytoplasmic (axial) resistivity. By definition, length constant = square_root ((radius/2) * R_m_ / R_a_). Thus, it should be one order of magnitude larger, and hence the electrotonic lengths smaller by the same factor.

Further, because the length constant is a multiplicative constant in the exponential, summation followed by division in their formula does not cancel out the effect of changing its scale. (The authors might want to consult Gouwens and Wilson, (2009); these authors fit their parameters in a compartmental model to reproduce their patch recording data.) In the present model, does the focality of inhibition remain robust when the value of the length constant is made more realistic as suggested here? If not, can the authors reconcile this discrepancy with their experimental findings?

[Editors' note: further revisions were suggested prior to acceptance, as described below.]

Thank you for submitting your article "Localized inhibition in the *Drosophila* mushroom body" for consideration by *eLife*. Your revised article has been reviewed Ronald Calabrese as the Senior Editor, Mani Ramaswami as the Reviewing Editor, and two reviewers. The following individuals involved in review of your submission have agreed to reveal their identity: Tamara Boto (Reviewer #2).

The reviewers have discussed the reviews with one another and the Reviewing Editor has drafted this decision to help you prepare a revised submission.

Summary:

The present study addresses for the first time the presence of spatially segregated GABAergic inhibition in the mushroom body, the main learning and memory center in the fly brain. Amin et al. perform a well needed analysis of the spatial dynamics of APL activity. The authors observe that, calcium fluxes can be observed in distinct subdomains of a widely arborizing, non-spiking, GABAergic neuron, and propose that such localised/compartmentalized inhibition can arise through local excitation and passive electrical responses in APL processes. Although the contribution of localised inhibition to behaviourally relevant computations or hypothesised functions of the APL in memory suppression, reversal learning or sparse odor coding, remain to be addressed, the observations made here are relevant not only for understanding sensory representation and learning in *Drosophila*, but also point the potential importance of segregated inhibition by interneurons in neuronal networks in general.

Essential revisions

Although these comments can be addressed through relatively minor changes to the text, we particularly request that the final conceptual point – on the relative roles of lateral and self inhibition be constructively addressed with appropriate, clarifying edits through the manuscript.

1) Please note in Figure 3, when referring to α prime and β prime neurons, there is a problem of compatibility and the symbols are not displayed properly.

2) Calcium influx: In their rebuttal (point 1b) the authors write: "Dendritic calcium in fly neurons can arise in large part from direct calcium influx through nAChRs, not necessarily from voltage-gated calcium channels, and thus need not reflect depolarization (Oertner et al., 2001)." This potentially explains attenuated proximal inhibition in the calyx in Figure 6C and Figure 7C. But why, then, does such calcium influx not reflect depolarization triggered by a puff of ATP?

3) Extent of inhibition by APL: In their rebuttal and revision, the authors argue that leaky expression of P2X2 in non-APL neurons results in overestimation of inhibition of KCs by APL. Additionally, they argue, inhibition of KCs by APL in the calyx will result in broader inhibition since the calyx houses KC dendrites. Inhibition of KC dendrites will also result in decreased activity in the axons in the horizontal and vertical lobes. These arguments are offered to explain the broader KC inhibition that they observed. However, even if we accept this explanation for results shown in Figure 7A2-C2 and 7A4-C4, it is still puzzling why KC inhibition by puffing GABA also was not as localized as expected (Figure 7A3-C3 and Figure 7A5-C5). The authors discuss this result (subsection “Local GABA application mimics the inhibitory effect of local APL stimulation”; Discussion section), speculating that the broad inhibition may be caused by wider network activity. Although this is possible, no evidence provided here supports this claim. Taken together, the evidence provided by the authors falls short of proving that inhibition from APL is compartmentalized in the MB.

4) Minimizing the role of lateral inhibition: In their rebuttal (point 2) the authors write: "…we do not claim that there is no, or insignificant, lateral inhibition. In fact, given lateral inhibition from 2000 KCs, the sum total of lateral inhibition onto any given KC is still stronger than its own self-inhibition." This is an important and clear statement that does not, but should, appear in the text, too. Instead, throughout the text the authors repeatedly emphasize the impact of local inhibition at the expense of lateral inhibition (i.e., "strong self-inhibition compared to other-inhibition" (Discussion section). Throughout the manuscript, the authors should present a more balanced view of local and lateral inhibition, which both appear to substantially affect the responses of KCs.

---

## [Author Response]

Essential revisions:(1) The most significant issue that needs to be addressed is the extent to which the APL neuron's response is localized and pertain to data in Figure 6 and Figure 7 and their interpretation. Based on Figure 6 and Figure 7, the authors argue that locally activated APL preferentially inhibits local KCs. Consistent with this, the inhibition effect appeared more pronounced in local KCs than distal KCs when APL was locally activated in the horizontal lobe and vertical lobe. However:(1a) In Figure 6, local activation of APL neuron leads to hyperpolarisation of KCs in other regions as well, in addition to the respective region where APL neuron was activated. This should be explained and discussed.

We discuss this issue in the Discussion section.

(1b) In Figure 6C (and Figure 7C), local activation of APL by ATP injection into the calyx leads to depolarisation, rather than inhibition of proximal KCs in calyx region, which could potentially be due to leaky expression of P2X2 channel in KCs (subsection “APL inhibits Kenyon cells mostly locally”). However, this seems unlikely: considering that ATP-evoked activity in KCs is comparable throughout the mushroom body in the negative control flies (Figure 6D), the leaky expression cannot explain the greater inhibition in distal KCs than proximal KCs. Assuming that the excitatory responses of KCs we see from the ATP injection on the calyx (APL) is due to leaky expression of P2X2, it still does not explain the inhibition at the vertical and horizontal lobes, which are away further away from the APL activation site. The inhibition at the vertical and horizontal lobes contradicts panels A and B. This discrepancy appears to contradict one of their main findings that spatially restricted APL activity mostly inhibits KCs locally and needs to be addressed.

We apologize for being unclear. First, to be clear, the KC responses recorded in the calyx and the lobes are not from distinct “distal KCs” and “proximal KCs”. These responses are from the same KCs, whose dendrites comprise the calyx, and axons comprise the lobes, respectively. As such, their processes are continuous from the calyx to the lobes (see Figure 8A, Figure 8—figure supplement 1).

Second, as the reviewers have noted, calcium does not necessarily directly correspond to depolarization. Dendritic calcium in fly neurons can arise in large part from direct calcium influx through nAChRs, not necessarily from voltage-gated calcium channels, and thus need not reflect depolarization (Oertner et al., 2001). Both P2X2 and nicotinic acetylcholine receptors (nAChRs) are calcium-permeable (Lima and Miesenböck, 2005; Oertner et al., 2001), so we argue that the leaky expression of P2X2 (whether in KCs or PNs) caused calcium influx in KC dendrites directly through P2X2 channels (if leaky P2X2 is in KCs) or nAChRs (if leaky P2X2 is in PNs). In UAS-P2X2-only negative controls, the calcium influx (plus additional sodium influx) through these channels depolarizes the KCs enough to spike (thus causing calcium influx throughout KC axons). However, in APL>P2X2 flies, this depolarization is counteracted by GABA locally released from APL evoked by ATP stimulation, which opens GABAA receptors and thereby opens a chloride conductance, which shunts away the depolarization and pushes the Kenyon cell’s membrane potential toward the reversal potential of chloride, which prevents the Kenyon cell from spiking. No spikes means no calcium influx in KC axons. Thus local APL stimulation in the calyx blocks spontaneous and odor-evoked calcium influx in Kenyon cell axons, even though it doesn’t block the local calcium influx in the calyx caused by leaky P2X2 expression. We have revised the text to explain this more clearly (subsection “APL inhibits Kenyon cells mostly locally”).

(1c) This apparent discrepancy is considered in the Discussion section, speculating that it could be caused by GABA receptors. Again, the observed discrepancy contradicts their main finding that differential spatial inhibition in KCs arises from local activity in APL and the spatial arrangement of KC-APL and APL-KC synapses. The authors need to revisit their explanation for why APL, when locally activated in the calyx, preferentially inhibits distal KCs rather than proximal ones.

The fourth paragraph of the Discussion section is unrelated to this issue and rather discuss how can it be that in some cases KCs are more strongly inhibited closer to the axon initial segment than far away from it, given that action potentials travel from the axon initial segment to the distal tip. Our answer on the point raised here is given above to reviewer comment 1b.

(1d) Subsection “Local GABA application mimics the inhibitory effect of local APL stimulation”: "strongest at the site of stimulation but still affecting Kenyon cells far from the site of stimulation, to a lesser extent." This statement appears to contradict the assertion that APL provides mainly local inhibition to KCs. The authors should moderate their claims throughout the text to make clear that APL provides both local and more global inhibition.

We have moderated our claims to better reflect that APL inhibits KCs strongest locally, but also has a weaker widespread effect:

“…activity in APL and APL’s inhibitory effect on Kenyon cells are spatially localized (the latter somewhat less so)…” (Abstract)

“…activity in APL and APL’s inhibitory effect on Kenyon cells are spatially restricted (though the latter somewhat less so)…” (Introduction)

“Local stimulation of the APL neuron in the lobes strongly reduced the baseline GCaMP6f signal in Kenyon cells locally, but also had a smaller, widespread effect.” (subsection “APL inhibits Kenyon cells mostly locally”)

“This local activity in APL translates into a spatially non-uniform inhibitory effect on Kenyon cells that is strongest locally and becomes weaker farther from the site of APL stimulation.” (Discussion section)

“…local activity in APL leads to stronger inhibition of Kenyon cells nearby than far away…” (Discussion section)

(1e) If additional experiments are needed to better establish the author's conclusions, then these should be described in the discussion (e.g. an analysis of the odor responses in MBON of neighboring/distal compartments could be important to clarify some conclusions and predictions of the local activation model).

We thank the reviewers for their suggestion. Accordingly, we have added a paragraph that discusses the implication of the APL neuron’s mainly local inhibitory effect on MBON odor responses (Discussion section).

2) The authors perform a thorough computational analysis of the connectomics data and suggest that each Kenyon cell inhibits itself more than others. This model would support the idea of a spatially restricted APL-mediated inhibition of Kenyon cells, and the authors concluded that ".our physiological measurements of localized activity of APL, combined with the spatial arrangements of KC-APL and APL-KC synapses, predict that Kenyon cells disproportionately inhibit themselves compared to other individual Kenyon cells" (Discussion section). This conclusion contradicts previous results by the same authors. Lin et al., (2014) published not only that the feedback inhibition is from all Kenyon cells to all Kenyon cells, but taken the specific case of the γ lobes of the mushroom bodies, Lin et al. showed the opposite result than the one expected in the present study: blocking synaptic release in γ lobes results in a decrease of the calcium signal of those same γ Kenyon cells, which contradicts the idea of γ KC strongly inhibiting γ Kenyon cells through APL.

We respectfully submit that the reviewer may have misinterpreted the Lin et al., 2014 results in the γ lobe. True, in flies expressing shibire in γ KCs, raising the temperature reduced odor-evoked Ca^2+^ influx in the γ lobe. However, this result should be attributed to the temperature increase alone, rather than shibire blocking synaptic vesicle release, because even in control flies that expressed no shibire, raising the temperature also reduced odor-evoked Ca^2+^ influx in the γ lobe (Figure 2 of Lin et al., 2014). Thus, those results actually suggest that blocking γ KC output has no effect on odor responses in the γ lobe.

These earlier results are perfectly consistent with our present results. While it is true that our model predicts that KCs disproportionately inhibit themselves more than other KCs, we do not claim that there is no, or insignificant, lateral inhibition. In fact, given lateral inhibition from 2000 KCs, the sum total of lateral inhibition onto any given KC is still stronger than its own self-inhibition. Thus, because γ KCs make up only ~1/3 of KCs, when output from γ KCs is blocked, the lateral inhibition from the remaining KCs should be sufficient to suppress odor responses in the γ lobe. Therefore, our present results do not contradict the finding in Lin et al., 2014 that feedback inhibition from the APL neuron onto KCs can operate in an all-to-all manner. We added a new paragraph in the Discussion section to address this point.

The manuscript should be revised to include a more thorough discussion regarding these contradictory results. The APL inhibition might vary depending on specific situations; all-to-all feedback might be essential for odor discrimination, whereas a more compartmentalized inhibition might be crucial during reinforcement: APL receives direct spatially-segregated input from dopaminergic neurons (Zhou et al., 2019; and Figure 1 of the present article) and these connections are not included in the connectomic analysis.

We completely agree, and discuss in the penultimate paragraph of the Discussion section how compartmentalized inhibition might contribute to learning. We added a specific mention of local dopaminergic input: “Thus, APL could locally modulate different compartment-specific aspects of olfactory learning, especially given that different regions of APL respond differently to dopamine (Zhou et al., 2019) and electric shock punishment (Figure 1)”. We may add dopaminergic inputs to our connectomic analysis in future studies.

(3) Further, regarding Figure 8F-L and connectome analysis in methods, the authors appear to make an error in their choice of length constant. The authors chose length constant = square_root(radius) (subsection “Connectome analysis”), leading them to a conclusion consistent with their experimental observation. However, published physiological data suggest that membrane resistivity should really be about two to three orders of magnitude higher than cytoplasmic (axial) resistivity. By definition, length constant = square_root ((radius/2) * R_m_ / R_a_). Thus, it should be one order of magnitude larger, and hence the electrotonic lengths smaller by the same factor.Further, because the length constant is a multiplicative constant in the exponential, summation followed by division in their formula does not cancel out the effect of changing its scale. (The authors might want to consult Gouwens and Wilson, (2009); these authors fit their parameters in a compartmental model to reproduce their patch recording data.) In the present model, does the focality of inhibition remain robust when the value of the length constant is made more realistic as suggested here? If not, can the authors reconcile this discrepancy with their experimental findings?

We apologize for not describing our methods more clearly. We did not literally set length constant = sqrt(radius), although we can see how our original Materials and methods section created that impression. Rather, length constant is *proportional* to sqrt(radius), and the whole point of our curve fitting in Figure 8E-F (was Figure 8D-E) was to figure out the constant of proportionality between the length constant and sqrt(r), which is determined biophysically by the factors the reviewer points out, i.e. the ratio of membrane to axial resistivity.

We have laid out the mathematical derivation of our distance calculations more extensively in the revised Materials and methods section. Note that we now treat skeleton segments not as cylinders but as truncated cones (see Scheffer et al., 2020), and we test a wide range of space constants derived from different ratios of membrane to axial resistivity (Figure 8—figure supplement 2; next text in Results section). Notably, the ratio of membrane to axial resistivity is much lower in a non-spiking interneuron in the visual system (HS cells) (Cuntz et al., 2013) compared to olfactory projection neurons (Gouwens and Wilson, 2009). We note that the ratio of membrane to axial resistivity that best fits our data is unusually low compared to the range of reported estimates of this ratio, and discuss this in subsection “Connectomic analysis predicts that each Kenyon cell disproportionately inhibits itself”. For example, if APL has active conductances that lower the membrane resistance when APL is depolarized, but not at rest, then its space constant would be shorter than would be predicted from purely passive properties. We also note caveats for the connectome-based model in Discussion section.

We now also report that if we treat all neurite segments as having the same radius (Figure 8—figure supplement 2), we obtain similar results to those in Figure 8 (subsection “Connectomic analysis predicts that each Kenyon cell disproportionately inhibits itself”).

[Editors' note: further revisions were suggested prior to acceptance, as described below.]

Essential revisions:[…]1) Please note in Figure 3, when referring to α prime and β prime neurons, there is a problem of compatibility and the symbols are not displayed properly.

Thank you – now fixed.

2) Calcium influx: In their rebuttal (point 1b) the authors write: "Dendritic calcium in fly neurons can arise in large part from direct calcium influx through nAChRs, not necessarily from voltage-gated calcium channels, and thus need not reflect depolarization (Oertner et al., 2001)." This potentially explains attenuated proximal inhibition in the calyx in Figure 6C and Figure 7C. But why, then, does such calcium influx not reflect depolarization triggered by a puff of ATP?

We are unsure what the reviewer is referring to here. Calcium influx does reflect depolarization triggered by a puff of ATP in Figure 4 and Figure 5. In Figure 6 and Figure 7, calcium influx does not reflect depolarization triggered by a puff of ATP because the neuron being depolarized by ATP (APL) is not the neuron whose calcium influx is being imaged (KCs). If this is about the leaky expression of P2X2 in Figure 6C and Figure 7C, we surmise that the reviewer is asking why is it that, in KC>GCaMP6f, APL>P2X2 flies, puffing ATP on the calyx increases calcium influx in KCs in the calyx but decreases calcium influx in the lobes. That is, why doesn’t leaky P2X2 expression in KCs cause depolarization (and thus calcium influx) in KC axons? The answer is that the ATP puff locally activates APL, which releases GABA onto the KCs, which counteracts the depolarization in KCs caused by leaky P2X2 expression (but still allows calcium influx directly through nAChRs or P2X2), and prevents the KCs from firing action potentials, thus blocking depolarization (and calcium influx) in KC axons. This explanation is supported by the fact that in negative control (UAS-P2X2 only) flies, in the absence of ATP-induced inhibition from APL, the ATP puff does increase calcium influx throughout KC axons. This reasoning is explained in subsection “APL inhibits Kenyon cells mostly locally”.

3) Extent of inhibition by APL: In their rebuttal and revision, the authors argue that leaky expression of P2X2 in non-APL neurons results in overestimation of inhibition of KCs by APL.

This is not the reviewer’s main point, but because this will be published with the main article if the manuscript is accepted, we would like to clarify that we argued that leaky expression of P2X2 in non-APL neurons *that excite APL could potentially* make us overestimate the *spatialextent* of *spread of activation within APL*, and we do not argue that this explains why inhibition by APL appears to spread more widely than activity within APL itself. (Discussion section).

Additionally, they argue, inhibition of KCs by APL in the calyx will result in broader inhibition since the calyx houses KC dendrites. Inhibition of KC dendrites will also result in decreased activity in the axons in the horizontal and vertical lobes. These arguments are offered to explain the broader KC inhibition that they observed. However, even if we accept this explanation for results shown in Figure 7A2-C2 and 7A4-C4, it is still puzzling why KC inhibition by puffing GABA also was not as localized as expected (Figure 7A3-C3 and Figure 7A5-C5). The authors discuss this result (subsection “Local GABA application mimics the inhibitory effect of local APL stimulation”; Discussion section), speculating that the broad inhibition may be caused by wider network activity. Although this is possible, no evidence provided here supports this claim. Taken together, the evidence provided by the authors falls short of proving that inhibition from APL is compartmentalized in the MB.

We also find this result puzzling and agree that our discussions of this result are indeed speculation. We have added “We speculate that…” and “Future experiments may test these possibilities” into the relevant paragraph of the Discussion section.

We have shown that inhibition from localized activation of APL in the MB lobes is, indeed, localized. Broader than the localized activation of APL itself, yes, but still localized. That is, when APL is locally activated in the lobes, KCs are more strongly inhibited close to the stimulation site, compared to far away. Thanks to the reviewers’ important comments on this point in the previous round of revision, the text is now appropriately nuanced on this point by stressing that although inhibition from APL is spatially non-uniform, it is somewhat less localized than the activation of APL itself.

4) Minimizing the role of lateral inhibition: In their rebuttal (point 2) the authors write: "…we do not claim that there is no, or insignificant, lateral inhibition. In fact, given lateral inhibition from 2000 KCs, the sum total of lateral inhibition onto any given KC is still stronger than its own self-inhibition." This is an important and clear statement that does not, but should, appear in the text, too. Instead, throughout the text the authors repeatedly emphasize the impact of local inhibition at the expense of lateral inhibition (i.e., "strong self-inhibition compared to other-inhibition" (Discussion section). Throughout the manuscript, the authors should present a more balanced view of local and lateral inhibition, which both appear to substantially affect the responses of KCs.

We thank the reviewer for this point. We agree and have added this statement into both the Results section and Discussion section. We also replaced ‘strong’ in the phrase mentioned by the reviewer with ‘disproportionate’ (Discussion section).

Elsewhere in the text, e.g., in the Abstract (“individual Kenyon cells inhibit themselves via APL more strongly than they inhibit other *individual* Kenyon cells”), we deliberately added the second “individual” (emphasis added) to indicate that we do not mean that self-inhibition in total is stronger than lateral inhibition in total, rather that the inhibition from the median KC1 onto itself is stronger than the average inhibition from KC1 onto an arbitrarily chosen single KC2.